# INSTRUCTMIX2MIX: CONSISTENT SPARSE-VIEW EDITING THROUGH MULTI-VIEW MODEL PERSONALIZATION

## ABSTRACT

We address the task of multi-view image editing from sparse input views, where the inputs can be seen as a *mix of images* capturing the scene from different viewpoints. The goal is to modify the scene according to a textual instruction while preserving consistency across all views. Existing methods, based on per-scene neural fields or temporal attention mechanisms, struggle in this setting, often producing artifacts and incoherent edits. We propose InstructMix2Mix (I-Mix2Mix), a framework that distills the editing capabilities of a 2D diffusion model into a pretrained multi-view diffusion model, leveraging its data-driven 3D prior for cross-view consistency. A key contribution is replacing the conventional neural field consolidator in Score Distillation Sampling (SDS) with a multi-view diffusion student, which requires novel adaptations: incremental student updates across timesteps, a specialized teacher noise scheduler to prevent degeneration, and an attention modification that enhances cross-view coherence without additional cost. Experiments demonstrate that I-Mix2Mix significantly improves multi-view consistency while maintaining high per-frame edit quality.

## 1 INTRODUCTION

Multi-view image editing seeks to modify a scene captured from multiple viewpoints while preserving consistency across views. Typical edits include texture and color changes, semantic manipulations, or geometric transformations, with applications in product imagery, real-estate and interior design visualization, AR/VR, and cinematic post-production.

However, multi-view editing is a difficult task that traditionally requires skilled artists, making automation highly desirable. In recent years, a variety of methods have been proposed for editing 3D scenes. Due to the difficulty of producing supervision in the form of high-quality paired scenes before and after editing, most of these approaches avoid direct multi-view editing, instead leveraging monocular editors such as InstructPix2Pix Brooks et al. (2023), either iteratively (Haque et al., 2023; Vachha & Haque, 2024) or through distillation (Li et al., 2024; Kamata et al., 2023). These approaches achieve 3D-consistent edits by operating on dense scene representations like NeRFs (Mildenhall et al., 2021) or 3D Gaussian Splats (Kerbl et al., 2023). However, in practice users often have only a sparse set of images, which provide limited scene coverage and cause existing methods to produce artifacts and inconsistent edits.

A complementary line of work, adapted from video editing, modifies the self-attention mechanism of 2D editors to encourage cross-frame coherence. While effective for semantic consistency, it struggles to preserve fine details under large viewpoint changes. Together, these challenges highlight the need for methods that enable robust, high-quality multi-view editing from sparse inputs.

In this work, we tackle the challenging problem of multi-view editing from sparse input images (a *mix* of images). Given a few source views and a textual editing instruction, our aim is to generate edits that faithfully follow the prompt while remaining consistent across all viewpoints. Following prior work, we leverage powerful 2D editors and lift their capabilities to 3D using Score Distillation Sampling (SDS) (Poole et al., 2022). However, we propose a new approach to this paradigm. We observe that current approaches face inherent limitations. Neural field representations are trained per scene – they do not hold 3D prior in their network weights. Instead, they achieve 3D consistency

by incorporating a physical prior through the rendering equations. A dense set of overlapping input images is required, however, to transform this prior into an effective consolidator. To achieve consistency with *sparse* views, we instead propose to incorporate a consolidator that embeds a strong, data-driven 3D prior directly in its weights: a multi-view synthesis diffusion model. While such models are trained to generate view-consistent scenes (e.g., from text or images), they lack editing capabilities. We bridge this gap by combining the strengths of both paradigms—distilling edits from a 2D editor (the teacher) into the multi-view model (the student). Concretely, we use InstructPix2Pix as the teacher and Stable Virtual Camera (SEVA) (Zhou et al., 2025) as the student. By leveraging a student model with an inherent 3D prior, our method — **I-Mix2Mix** — produces robust, geometrically coherent, and visually consistent edits even from extremely sparse inputs.

Replacing the neural field with a multi-view diffusion model within the SDS framework is not straightforward, and requires careful adaptation of several key steps. Instead of rendering from a scene representation, we sample from the diffusion model; to avoid costly full trajectories, we distill incrementally across student timesteps. We also introduce a specialized teacher noise scheduler to prevent collapse to poor local minima and an attention modification that strengthens multi-view consistency without extra cost. Together, these components yield a framework for consistent multi-view editing, producing high-quality results even with very sparse inputs.

To summarize, our contributions are:

1. We present I-Mix2Mix, a novel framework for distilling the knowledge of a powerful monocular editor into a pretrained multi-view diffusion model, leveraging its data-driven 3D consistency.

2. Through careful consideration of the SDS key steps, we introduce novel adaptations to support personalization of our multi-view student.

3. We demonstrate that this approach produces high-quality, consistent multi-view edits, effectively extending the SDS framework to scenarios with limited viewpoints.

We evaluate I-Mix2Mix against popular multi-view editing methods, demonstrating significant improvements in cross-view consistency both qualitatively and quantitatively in the sparse-view setting. At the same time, our method maintains competitive per-frame editing performance, highlighting the practical benefits of leveraging a data-driven multi-view prior within the SDS framework.

## 2 RELATED WORKS

**Neural field editing.** Editing 3D scenes or objects typically assumes a pre-optimized model such as a NeRF (Mildenhall et al., 2021) or 3D Gaussian Splatting (Kerbl et al., 2023). Early works explored *direct NeRF manipulation* via scribbles (Liu et al., 2021), sketches (Mikaeili et al., 2023), reference images (Bao et al., 2023), meshes (Yuan et al., 2022), point clouds (Chen et al., 2023a), and other cues (Weder et al., 2023; Mirzaei et al., 2023; Yang et al., 2021), while *NeRF stylization* transfers reference appearances to 3D scenes (Wang et al., 2022; 2023a; Nguyen-Phuoc et al., 2022; Huang et al., 2022; Chiang et al., 2022). *Instruction-based* approaches leverage 2D diffusion editors like InstructPix2Pix (Brooks et al., 2023), applying SDS-like guidance (Li et al., 2024; Sella et al., 2023; Zhuang et al., 2023; Kamata et al., 2023) or Iterative Dataset Update (Haque et al., 2023; Wang et al., 2024a; Vachha & Haque, 2024; Wang et al., 2024b; Chen et al., 2024c;a) for improved consistency. Distinctively, SHAP-Editor (Chen et al., 2023b) operates in latent space for feed-forward edits. While effective for 3D editing, the reliance of these approaches on dense input views makes them less suitable in sparse-view scenarios.

**Sparse multi-view editing.** In the absence of a full 3D representation, several works have explored editing a set of input images directly. A prominent direction adapts pre-trained diffusion-based monocular editors by modifying self-attention layers: as first shown in Wu et al. (2023), extending queries to attend across frames promotes consistency between the outputs. Building on this idea, a number of methods generate *edited videos* (Geyer et al., 2023; Shin et al., 2024; Khachatryan et al., 2023; Ceylan et al., 2023; Qi et al., 2023; Liu et al., 2024). While effective for temporally smooth sequences with small viewpoint changes, these approaches struggle in the sparse-view setting, where edits must remain consistent under significant viewpoint differences. DGE (Chen et al., 2024b) combines extended attention with 3D Gaussian Splatting (3DGS) lifting: attention-based edits provide rough multi-view consistency, while 3DGS is used to consolidate outputs and resolve residual

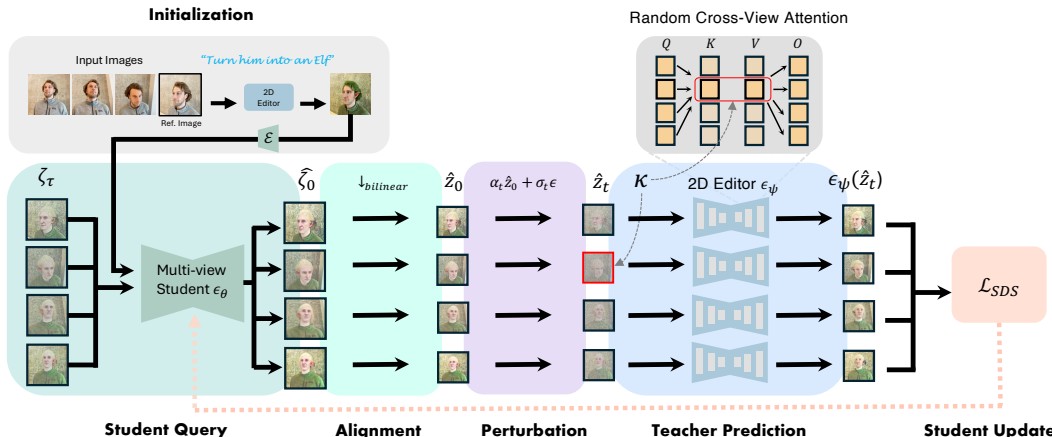

Figure 1: **I-Mix2Mix overview**. Given a set of input images, a randomly chosen reference image is edited by the frozen teacher and encoded to serve as the personalized multi-view student's input latent (*Initialization*). At each distillation iteration, noisy multi-view latents $\zeta_\tau$ are denoised by the student (*Student Query*), aligned to the teacher's latent space (*Alignment*), and perturbed with our forward schedule (*Perturbation*). The teacher predicts edits with Random Cross-View Attention (*Teacher Prediction*), where all frames attend to the $\kappa$'s frame, and the resulting supervision is distilled back into the student (*Student Update*). After distillation, the student outputs a set of multi-view consistent edited frames.

artifacts. However, in the sparse-view regime, 3DGS tends to overfit the limited input views rather than serve as a true cross-view aggregator, leading to persistent inconsistencies. As a result, DGE effectively reduces to an extended-attention approach, inheriting the same limitations as prior video editing methods. Most recently, Bar-On et al. (2025) proposed a feed-forward approach that propagates a user-provided 2D edit to multiple views, but their method remains limited to object-level edits. Contemporaneously with our work, Zhao et al. (2025) fine-tune FLUX Kontext (Labs et al., 2025) to enable consistent edits across image pairs, while Chi et al. (2025) distill 3D consistency priors into a 2D editor through a VSD-based framework (Wang et al., 2023b).

## 3 PRELIMINARIES

**Stable Virtual Camera.** SEVA (Zhou et al., 2025) is a diffusion-based Novel View Synthesis (NVS) model that predicts $N$ target images given $M$ input images with their camera poses. Built on Stable Diffusion 2.1 Rombach et al. (2022) with architectural adaptations for NVS and trained on diverse object and scene datasets, it achieves state-of-the-art results, making it an ideal student model with a strong 3D prior.

**Instruct-Pix2Pix.** A monocular, instruction-based image editing diffusion model widely used in 3D and multi-view editing, Instruct-Pix2Pix (Brooks et al., 2023) is fine-tuned from a pre-trained Stable Diffusion model on a large-scale synthetic editing dataset. Given a source image and a textual instruction, it produces versatile edits by sampling the fine-tuned model. The model employs classifier-free guidance (CFG) (Ho & Salimans, 2022) with two scales: a *text CFG scale* $s_T$ controlling adherence to the instruction, and an *image CFG scale* $s_I$ controlling fidelity to the source image, jointly balancing edit strength and overall image quality.

## 4 METHOD

Our goal is sparse multi-view consistent image editing. We build on the SDS framework (Poole et al., 2022), using a pre-trained image editing network as a *teacher* to distill knowledge into a neural scene representation *student*. Unlike typical settings that assume abundant input views, we work with only a few images. To address this challenge, we replace the conventional neural field with a multi-view diffusion model pre-trained for consistent view generation. We personalize this

student to the target scene and edit instruction by distilling the teacher's predictions, enabling faithful and consistent edits from limited inputs.

## 4.1 PROBLEM FORMULATION

We are given $N$ images $\{I_i\}_{i=1}^N$, $I_i \in \mathbb{R}^{3 \times H \times W}$ of a static 3D scene with camera poses $\{\pi_i\}_{i=1}^N$, $\pi_i \in \mathbb{R}^{4 \times 4}$, and an editing prompt $y \in \mathcal{Y}$. We assume access to a multi-view diffusion model $\epsilon_\theta$ which we refer to as the *student*, and a monocular instruction-based editing diffusion model (*teacher*) $\epsilon_\psi$. The goal is to produce edited views $\{E_i\}_{i=1}^N$ such that (i) each $E_i$ is a faithful edit of $I_i$ according to $y$, and (ii) $\{E_i\}$ are multi-view consistent, i.e. there exists a underlying 3D scene representation $\mathcal{S}$ whose renderings under poses $\{\pi_i\}$ yield $\{E_i\}$.

## 4.2 SCORE DISTILLATION SAMPLING

Originally introduced in *DreamFusion* (Poole et al., 2022) for 3D generation using 2D diffusion models, **Score Distillation Sampling (SDS)** is an iterative technique for utilizing the generative prior of a pre-trained diffusion model $\epsilon_\psi$ (*teacher*) to tune the parameters $\theta$ of a differentiable neural scene representation $\Phi_\theta$ (*student*). At each iteration, the student is queried (rendered) through a differentiable operator $g$, yielding $\hat{\chi}_0 = g(\Phi_\theta)$. This prediction is then critiqued by the teacher, and the process repeats iteratively, updating $\theta$ until the student encodes a scene representation $\Phi_\theta$ that yields plausible renderings. The overall SDS framework, can be summarized as a five-stage pipeline, schematically shown in the inset figure:

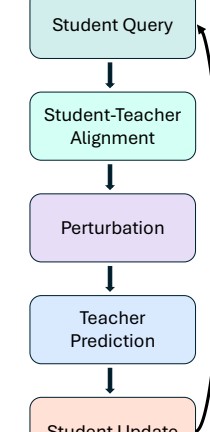

**1. Student Query.** The student $\Phi_\theta$ produces an image or latent $\hat{\chi}_0 = g(\Phi_\theta)$ to be critiqued. Commonly this is a differentiable rendering from a NeRF.

**2. Student-Teacher Alignment.** The student output $\hat{\chi}_0$ is mapped to the teacher's input space as $\hat{x}_0$, e.g., through an image encoder.

**3. Perturbation.** The aligned prediction $\hat{x}_0$ is perturbed according to the teacher's forward diffusion process: $\hat{x}_t = \alpha_t \hat{x}_0 + \sigma_t \epsilon$, $\epsilon \sim \mathcal{N}(0, \boldsymbol{I})$.

**4. Teacher Prediction.** The teacher model $\epsilon_\psi$ processes $\hat{x}_t$ (conditioned on an embedding $y$ and the timestep $t$), and predicts the corresponding noise $\epsilon_\psi(\hat{x}_t; y, t)$.

**5. Student Update.** The residual between the sampled noise $\epsilon$ and the teacher's prediction $\epsilon_\psi(\hat{x}_t; y, t)$ defines the SDS gradient: $\nabla_\theta \mathcal{L}_{\text{SDS}} = \mathbb{E}_{t,\epsilon} \left[ w(t) \left( \epsilon_\psi(\hat{x}_t; y, t) - \epsilon \right) \frac{\partial \hat{x}_0}{\partial \theta} \right]$, where $w(t)$ is a time-dependent weighting term. The gradient is backpropagated to update the student parameters $\theta$.

SDS variants differ primarily in the specific design choice at each stage.

## 4.3 CONSISTENT SPARSE-VIEW EDITING THROUGH STUDENT PERSONALIZATION

In our framework, where the student is a diffusion-based multi view synthesis model, key SDS stages require specialized adaptations, which we detail in this subsection. Our proposed approach is illustrated in Figure 1.

**Step 1: Student Query.** In traditional SDS with NeRFs or 3DGS, the student prediction $\hat{\chi}_0 = g(\Phi_\theta)$ is obtained via differentiable rendering. In our setting, the student is a multi-view diffusion model $\epsilon_\theta$, so the analogue is generating a sample via its denoising trajectory (Ho et al., 2020; Song et al., 2020). Running a full sampling trajectory at each SDS iteration is however slow and computationally expensive, requiring backpropagation through many denoiser evaluations. Instead, we distill incrementally at each student timestep $\tau$, starting from $\tau = \text{T}$ with latents sampled from the Gaussian distribution with student-scheduler specified variance $\{\zeta_\text{T}^i\}_{i=1}^N \sim \mathcal{N}(0, \sigma_\text{S}^2 \boldsymbol{I})$. We compute *single-step predictions* of the clean latents via the Tweedie formula (Efron, 2011). These estimates $\{\hat{\zeta}_0^i(\tau)\}$ serve as intermediate student predictions to be critiqued by the teacher, shaping the student's backward trajectory step by step.

**Step 2: Student-Teacher Alignment.**  Although both student and teacher are latent diffusion models, they operate in different latent spaces and dimensions. A naive approach would decode the student's predictions $\{\hat{\zeta}_0^i\}$ with the its decoder $\mathcal{D}_S$ and encode them with the teacher encoder $\mathcal{E}_T$ before adding noise via the teacher's forward process. However, backpropagating through both $\mathcal{D}_S$ and $\mathcal{E}_T$ would be prohibitively expensive. Inspired by prior work on convergent representations (Asperti & Tonelli, 2023; Lenc & Vedaldi, 2015; Huh et al., 2024; Li et al., 2015), which suggests that simple mappings can often bridge the representation spaces of different networks, we instead resize the student's latents to the teacher's expected dimensions $(H_T, W_T)$ via bilinear interpolation: $\hat{z}_0^i = \mathcal{I}_{bilinear}(\hat{\zeta}_0^i; H_T, W_T)$. For our chosen models, this lightweight approach suffices, suggesting that the student latents implicitly align with the teacher's latent space during fine-tuning.

**Step 3: Perturbation.**  The mapped latents are perturbed using the teacher's forward process $\hat{z}_t^i = \alpha_t \hat{z}_0^i + \sigma_t \epsilon_i, \ \epsilon_i \sim \mathcal{N}(0, \boldsymbol{I})$, yielding noisy latents $\{\hat{z}_t^i\}$. A key design choice is the teacher timestep $t$. In standard SDS (Poole et al., 2022), $t$ is drawn uniformly from $[0.02, 0.98]$, avoiding extreme noise levels for numerical stability. This is ill-suited to our setting: early student outputs (large $\tau$) lie off the natural image manifold, so at low $t$ values their diffused versions fall outside the teacher's distribution, causing unstable guidance.

Annealed $t$ schedules have also been explored (Huang et al., 2023; Lukoianov et al., 2024), and a natural variant is to match $t$ with the student timestep $\tau$. Yet this is too restrictive—when $\tau$ is small, forcing $t \approx \tau$ limits the teacher's ability to provide corrective gradients. We instead use a stochastic schedule: $t \sim \text{TruncNorm}\left(\mu = b, \ \sigma = \frac{b-\tau}{f}, \ a = \tau, \ b = 0.95\right)$, where $f$ controls skewness. Larger $f$ concentrates probability near $b$, making it more likely for the teacher to operate at higher noise levels. The randomness ensures that the teacher provides strong gradients every few iterations, which we find highly effective for avoiding collapse to poor local minima. See Appendix A.1 for further details and visualizations.

**Step 4: Teacher Prediction.**  A straightforward application of our framework would pass the perturbed latents $\{\hat{z}_t^i\}_{i=1}^N$ as a batch to the monocular teacher U-Net $\epsilon_\psi$, which would then produce independent noise estimates for each latent. Backpropagating such conflicting signals into the student can weaken its multi-view prior, yielding inconsistent final edits. To address this, we introduce a lightweight Random Cross-View Attention (RCVAttn) mechanism that encourages the teacher to generate more consistent edits within each batch. Inspired by attention-based alignment work (Khachatryan et al., 2023), we randomly select a *key frame* index $\kappa \sim U\{1, \ldots, N\}$ at each iteration. Each frame $i$ attends to the tokens of the key frame:

$$\text{RCVAttn}(Q, K, V, i) = \text{softmax}\left(\frac{Q_i K_\kappa^\top}{\sqrt{d}}\right) V_\kappa, \tag{1}$$

where $d$ is the query/key dimensionality. Aligning all frames to query the key frame improves consistency substantially, aiding in retaining the student's multi-view prior. Unlike expensive extended-attention methods (Wu et al., 2023; Chen et al., 2024b; Geyer et al., 2023), RC-VAttn adds no computational overhead. While non-key frames may experience reduced quality, randomly selecting $\kappa$ ensures all frames occasionally serve as the key, preventing noticeable degradation. The effect of RCVAttn, when applied to full teacher sampling process, is shown in the inset figure.

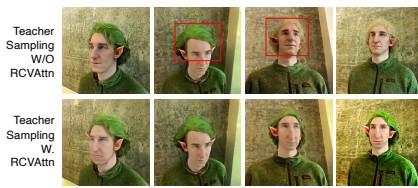

**Step 5: Student Update.**  Finally, the difference between the sampled noise and the teacher's prediction defines the guidance direction in the SDS objective:

$$\nabla_\theta \mathcal{L}_{\text{SDS}} = \frac{1}{N} \sum_{i=1}^N \left(\epsilon_\psi(\hat{z}_t^i; y, I_i, t) - \epsilon_i\right) \frac{\partial \hat{z}_0^i}{\partial \theta}, \tag{2}$$

which is backpropagated to update the student weights.

This completes a single distillation iteration at student timestep $\tau$. We start at $\tau = \text{T}$ and repeat this process for $k$ iterations, personalizing the student model at this timestep to the indented edit. The student then performs a sampling step with its scheduler, producing latents $\{\zeta_{\tau-\Delta\tau}^i\}$, where $\Delta\tau$ is the step size. Distillation resumes at $\tau - \Delta\tau$, repeating $k$ updates before the next sampling step. This nested procedure continues until $\tau = 0$, yielding final edited views that are instruction-faithful and multi-view consistent.

**Initialization.** Our student model, SEVA, is an "$M$ in, $N$ out" model with $M \geq 1$, meaning that the denoiser $\epsilon_\theta$ requires at least one clean input latent in addition to the $N$ noisy latents. As a preprocessing step, we randomly select one of the input frames $I_{\text{ref}} \in \{I_i\}$ and pass it through the 2D teacher editor with prompt $y$ to generate a valid reference edit $E_{\text{ref}}$. This edit is then encoded using SEVA's frozen encoder to obtain a reference latent $z_{\text{ref}} = \mathcal{E}_{\text{S}}(E_{\text{ref}})$, which serves as the input frame to the denoiser in all distillation iterations. This can be considered "Step 0" of the framework. The full framework is summarized in the inset algorithm.

---

**Algorithm 1: I-Mix2Mix**

1: **Input:**
2: $\quad \{I_i\}_{i=1}^N, \{\pi_i\}_{i=1}^N, y$    ▷ Images, poses and text prompt
3: $\quad \epsilon_\psi, \epsilon_\theta$    ▷ Frozen teacher and trainable student
4: $z_{\text{ref}} \leftarrow \mathcal{E}_S(\text{TeacherEdit}(I_{\text{ref}}, y))$
5: Initialize $\zeta_T^i \sim \mathcal{N}(0, \sigma_S^2 I)$
6: **for** $\tau = T, T - \Delta\tau, \ldots, 0$ **do**
7:    **for** $k$ steps **do**
8:       $\{\hat{\zeta}_0^i\} \leftarrow \epsilon_\theta(\{\zeta_\tau^i\}, z_{\text{ref}}, \{\pi_i\})$
9:       $\hat{z}_0^i \leftarrow \mathcal{I}_{\text{bilinear}}(\hat{\zeta}_0^i)$
10:       $t \sim \text{TruncNorm}(\mu=b, \sigma=\frac{b-\tau}{f}, a=\tau, b=0.95)$
11:       $\epsilon_i \sim \mathcal{N}(0, I)$
12:       $\hat{z}_t^i \leftarrow \alpha_t \hat{z}_0^i + \sigma_t \epsilon_i$
13:       $\kappa \sim U\{1, \ldots, N\}$    ▷ Select keyframe
14:       $\{\tilde{\epsilon}_i\} \leftarrow \epsilon_\psi(\{\hat{z}_t^i\}; y, \{I_i\}, t)$ ▷ With RCVAttn
15:       $\nabla_\theta \mathcal{L}_{\text{SDS}} \leftarrow \frac{1}{N} \sum_i (\tilde{\epsilon}_i - \epsilon_i) \frac{\partial \hat{z}_0^i}{\partial \theta}$
16:       $\theta \leftarrow \theta - \eta \nabla_\theta \mathcal{L}_{\text{SDS}}$
17:    **end for**
18:    $\{\zeta_{\tau-\Delta\tau}^i\} \leftarrow \text{StudentStep}(\{\zeta_\tau^i\}, z_{\text{ref}}, \{\pi_i\}, \Delta\tau)$
19: **end for**
20: **Output:** $E_i \leftarrow \mathcal{D}_T(\hat{z}_0^i)$

---

## 5 EXPERIMENTS

**Methods in comparison.** We compare with four widely used, open-source methods that also employ InstructPix2Pix as the 2D editor, covering distinct paradigms for multi-view editing: *Instruct-NeRF2NeRF (I-N2N)* (Haque et al., 2023) and its 3DGS variant *Instruct-GS2GS (I-GS2GS)* (Vachha & Haque, 2024) (both following the Iterative Dataset Update paradigm), *Text2Video-Zero (T2VZ)* (Khachatryan et al., 2023) (a zero-shot image-to-video adaptation), and *DGE* (Chen et al., 2024b) (extended attention for multi-view editing with 3DGS-based consolidation). Since I-N2N requires a trained NeRF, we optimize a Nerfacto (Tancik et al., 2023) model on the $N$ input views; similarly, because I-GS2GS and DGE require a 3DGS, we optimize a Splatfacto (Tancik et al., 2023) model. All baselines are run with default settings from the official implementations or papers.

**Evaluation.** We evaluate our method on scenes from several datasets: I-N2N (Haque et al., 2023), Tanks and Temples (Knapitsch et al., 2017), CO3D (Reizenstein et al., 2021), and Mip-NeRF 360 (Barron et al., 2022). Following prior protocols, for comparison with baselines we apply 20 edits to three standard test scenes from I-N2N (full edit set detailed in Appendix B); qualitative results on additional scenes appear in Appendix F. Per-frame edit quality and cross-view consistency are assessed with three CLIP-based (Radford et al., 2021) metrics commonly used in prior work (Haque et al., 2023; Chi et al., 2025; Chen et al., 2024b): (i) *CLIP Similarity*, the cosine similarity between an edited image and the prompt; (ii) *CLIP Directional Similarity* (Gal et al., 2022; Brooks et al., 2023), which measures alignment between prompt change and image change; (iii) *CLIP Directional Consistency* (Haque et al., 2023), which quantifies multi-view consistency by comparing the relative changes between pairs of original views $O_i, O_j$ and their corresponding edited views $E_i, E_j$ via $cos\_sim\big(\phi(O_i) - \phi(O_j), \ \phi(E_i) - \phi(E_j)\big)$, where $\phi(\cdot)$ denotes the CLIP embedding. This metric captures whether the semantic difference between two views is preserved after editing. Unlike the original formulation, which considers only consecutive frames, we average over all $\binom{N}{2}$ pairs to account for our unordered, sparse-view setting.

We use $N = 4$ frames in main experiments, with additional results for larger $N$ in Appendix G. Full implementation details, are detailed in Appendix A.

### 5.1 COMPARISON WITH PRIOR WORK

Quantitative results are reported in Table 1, and representative qualitative comparisons are shown in Figure 2. Enlarged visualizations and additional examples are included in Appendix E.

Our method achieves the highest performance in *CLIP Directional Consistency (CLIP Cons.)*, indicating that edits remain more consistent across different views. Importantly, this does not come at

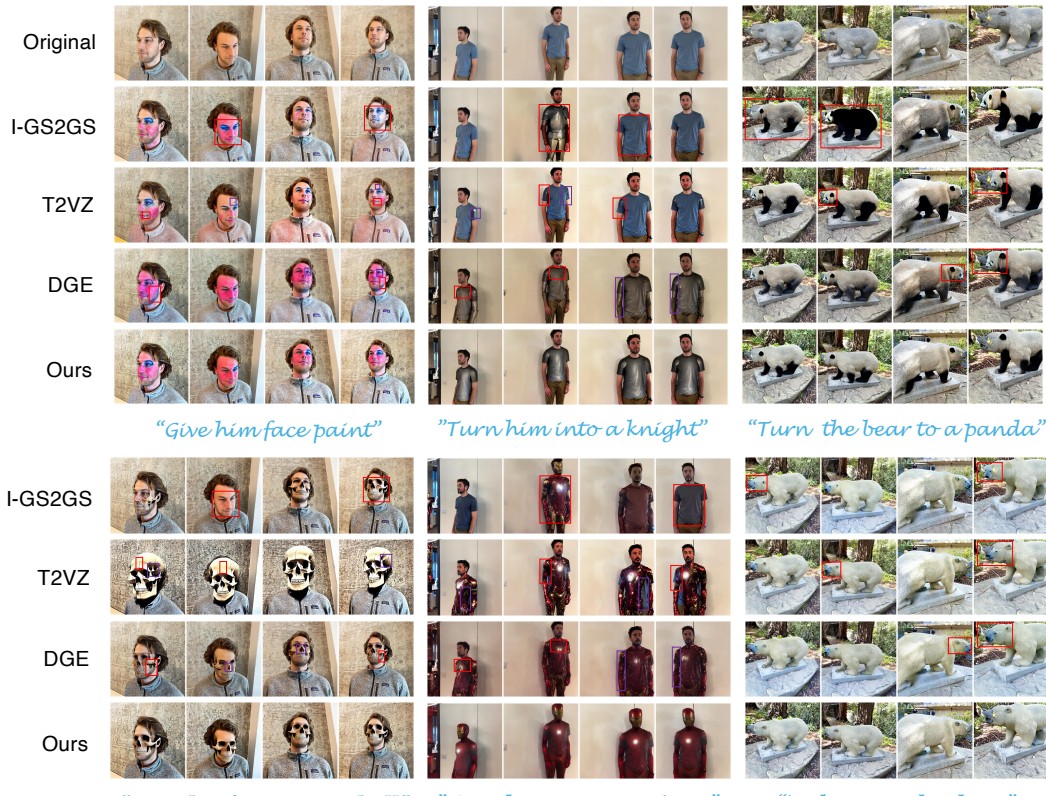

Figure 2: Qualitative comparison with prior work. The top row shows the original scenes, and the lower rows present edits from different methods. Matching red or purple rectangles indicate pairs of inconsistent regions, which frequently appear in baselines but not in our edits. Please zoom in electronically for details; enlarged views are provided in Appendix E.

| Method | CLIP Cons. ↑ | CLIP Sim. ↑ | CLIP Dir. ↑ |
|---|---|---|---|
| I-N2N | 0.034 | 0.196 | 0.105 |
| I-GS2GS | 0.314 | 0.253 | 0.169 |
| T2VZ | 0.310 | 0.251 | 0.159 |
| DGE | 0.287 | 0.256 | **0.182** |
| **Ours** | **0.342** | **0.258** | 0.173 |

Table 1: Comparison of methods across view consistency, semantic alignment and edit performance.

the cost of per-frame edit quality, as demonstrated by CLIP Sim. and CLIP Dir. scores, where our method is either superior to or competitive with the baselines.

Notably, I-N2N fails completely in this sparse-view setting. We observed that Nerfacto struggles to fit the scene, producing severe floater artifacts even when rendering source poses. These distortions lie out of distribution for the 2D editor, leading to unusable edits as shown in Appendix C.

The advantages of our approach compared to other baselines, are most clearly demonstrated qualitatively. In the sparse-view setting, baseline methods often struggle to maintain consistent edits across viewpoints due to two factors: (i) 3DGS-based consolidation becomes unreliable with limited views, as the 3DGS tends to overfit the training images; and (ii) cross-frame attention, while improving general appearance alignment, fails to enforce fine-grained consistency. Figure 2 illustrates these issues: I-GS2GS edits remain largely view-independent (e.g., the *Face Paint* edit), while T2VZ and DGE, though producing roughly appearance-consistent edits, introduce inconsistency in the details—such as mismatched sleeve and chest textures in the *Knight* and *Iron Man* edits, varying

| SDS Stage | Config | CLIP Cons. ↑ | CLIP Sim. ↑ | CLIP Dir. ↑ |
|---|---|---|---|---|
| – | Student Only | 0.014 | 0.212 | 0.161 |
| – | Teacher Only | 0.228 | 0.252 | 0.184 |
| Initialization (0) | W/O Editing Ref. Frame | 0.326 | 0.264 | 0.174 |
| Alignment (2) | Learned Mapping | 0.287 | 0.259 | 0.180 |
| Perturbation (3) | Uniform $t$ | 0.363 | 0.260 | 0.146 |
|  | $\tau$-matched $t$ | 0.435 | 0.231 | 0.107 |
| Teacher Pred. (4) | W/O RCVAttn | 0.230 | 0.260 | 0.175 |
|  | **Full** | 0.337 | 0.263 | 0.178 |

Table 2: Ablation study evaluating different design choices. Weak results are highlighted in red.

face paint colors and intensities in *Face Paint*, and view-dependent differences in *Skull* details such as the nose, cheek, and forehead. The lack of robust 3D consistency is especially evident in the *Bear* edit, which exhibits Janus-like multi-face artifacts. In contrast, I-Mix2Mix produces edits that are not only faithful to the instruction but also highly consistent across all views, without sacrificing image quality. This combination of instruction alignment, visual fidelity, and strong 3D consistency represents a clear improvement over existing approaches in the sparse-view setting.

## 5.2 ABLATION STUDY

We conduct an ablation study on 6 representative edits (listed in Appendix B) to assess the contributions of our design choices across the SDS pipeline. Quantitative results are summarized in Table 2, with particularly weak results highlighted in red. Our findings show that each component is essential to achieve both faithful edits and strong multi-view consistency.

**Role of teacher and student.** We first test the student and teacher models in isolation. In the *Student Only* setting, the student is given an edited frame as input and asked to sample additional views. This fails for several reasons: the student never sees the scene content captured by the other frames, leading to poor faithfulness; the SEVA prior struggles under single-view input; and we suspect that the edited scenes lie out-of-distribution for the model. This suggests that our approach *distills new capabilities into the student*, rather than simply searching within its existing sampling distribution. Conversely, in the *Teacher Only* setting, we rely on the teacher to edit each view independently. While individual frames adhere to the instruction edit, the lack of a 3D prior leads to severe cross-view inconsistency, as reflected by the low CLIP Consistency score. We present representative visualizations in Appendix D. Together, these results confirm the necessity of distilling from the teacher into the student, rather than using either in isolation.

**Initialization stage.** In the *W/O Editing Ref. Frame* setting, we input one of the original frames to the student encoder, to serve as the reference latent, without editing it first: $z_{\text{ref}} = \mathcal{E}_S(I_{\text{ref}})$. Skipping the reference frame edit negatively affected the distillation process, as the initial student predictions are further away from the target. This results in slightly lower multi-view consistency.

**Alignment stage.** Following findings in prior work on latent space alignment (Huh et al., 2024; Li et al., 2015), we replaced the bilinear interpolation with a learnable convolutional mapping (*Learned Mapping*), optimized during distillation, but found this brought no measurable benefit – the necessary transformation is likely captured during the fine-tuning of the student.



*"Turn him into a knight"*

**Perturbation stage.** We evaluated two alternatives to our proposed forward schedule: *Uniform $t$* (similar to Poole et al. (2022)), where $t$ is sampled uniformly in $[0.05, 0.95]$, and $\tau$-*matched $t$*, where the teacher's timestep follows the student's. Both variants tend to collapse to near-identity reconstructions of the input scene, which explains their paradoxically high CLIP Consistency: such outputs are trivially consistent but fail to realize

the intended edit as reflected in their low CLIP Directional scores. Visual examples are provided in the first two rows of the *Knight* inset figure.

**Teacher prediction stage.** Finally, disabling our Random Cross-View Attention mechanism (*W/O RCVAttn*) leads the teacher to process each perturbed latent independently. Without cross-view coupling, the student receives conflicting signals across views, leading to degraded multi-view consistency and breaking its 3D prior. This is again reflected in low CLIP Consistency, and illustrated in the third row of the *Knight* inset figure.

**Efficiency considerations.** Several components of I-Mix2Mix were explicitly designed for memory and compute efficiency. Our choice of using a single-step prediction in the *Student Query* stage is crucial: a three-step alternative more than doubles peak memory usage for $N = 4$ views 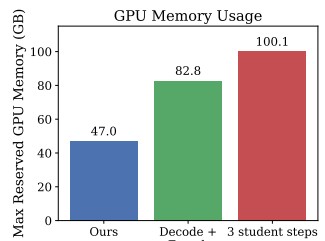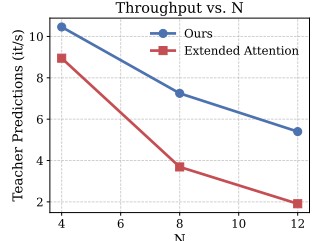
(see inset). In the *Student-Teacher Alignment* stage, interpolating latents rather than passing through the student's decoder and teacher's encoder reduces memory usage by over $50\%$. Finally, replacing the RCVAttn module with full extended attention significantly degraded throughput, worsening as the number of frames increased. We additionally experimented with fine-tuning the student using LoRA (Hu et al., 2022) rather than updating the full U-Net. While more parameter-efficient, this approach underperformed, and we leave the adaptation of lightweight variants to future work.

### 5.3 BEYOND IMAGE EDITING

Our approach is not inherently limited to editing tasks. In principle, any pre-trained image-to-image diffusion model can serve as the teacher, with the multi-view student acting as a consolidator to produce a multi-view–to–multi-view solution. To explore this, we experimented with multi-view conditional generation. Specifically, we employed pre-trained ControlNets (Zhang et al., 2023) as teachers to translate multiple depth maps or Canny edge maps of a 3D scene into consistent RGB images. Qualitative examples are provided in Appendix H. While the outputs were faithful to the conditioning inputs and maintained multi-view consistency, they often exhibited excessive blurriness—a known artifact of SDS-based optimization (Poole et al., 2022).

## 6 DISCUSSION: PARALLEL TO DIFFUSION GUIDANCE

In standard diffusion guidance (Bansal et al., 2023; Chung et al., 2022), the model predictions at a given timestep are often critiqued, and the resulting gradient is used to modify the sampling trajectory. In our framework, rather than applying such potentially unstable updates to the latents, we backpropagate the guidance signal to the student's weights. This approach effectively transfers the teacher's knowledge without making aggressive modifications to the latents themselves, avoiding manifold slips – divergence from the target distribution. Consequently, the student gradually learns to generate multi-view consistent edits while maintaining stable sampling dynamics.

## 7 CONCLUSION, LIMITATIONS, AND FUTURE WORK

We presented I-Mix2Mix, a novel framework for multi-view image editing that achieves high multi-view consistency in sparse-view settings, where prior methods typically fail. While effective, our approach inherits the limitations of its backbones, specifically InstructPix2Pix and SEVA, which can struggle with certain edit prompts or with maintaining perfect consistency across views. Given our framework's modular nature, we anticipate that integrating stronger future backbones could mitigate these issues. Additionally, I-Mix2Mix requires multiple distillation iterations per noise level, making it more than twice as slow as our strongest competitor, DGE. We plan to explore strategies to reduce this overhead in future work. Finally, as discussed in Section 5.3, our framework is potentially general and applicable to a range of image manipulation tasks beyond editing. However, performance on these tasks currently lags behind our editing results, often producing blurry outputs. We leave the investigation of these directions to future work.

**Ethics statement.** Our work builds on publicly available datasets (Haque et al., 2023; Knapitsch et al., 2017; Reizenstein et al., 2021; Barron et al., 2022) that were released for research purposes. Some of these datasets include human subjects, for which consent has been obtained as described in the original publications. As our method enables multi-view image editing, it could in principle be misused for generating misleading or harmful content (e.g., deepfakes). While our focus is on advancing controllable and consistent scene editing for academic and scientific applications, we caution that such techniques should be applied responsibly, in accordance with ethical standards and legal regulations.

**Reproducibility statement.** We provide a detailed description of our method in Section 4, along with a pseudo-algorithm in Algorithm 1. Implementation details are given in Appendix A. The evaluation protocol, datasets, and editing prompts are described in Section 5 and Appendix B.

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

APPENDIX

CONTENTS

## A  IMPLEMENTATION DETAILS

We use SEVA 1.1 (Zhou et al., 2025) as the pre-trained student model and InstructPix2Pix (Brooks et al., 2023) from the Diffusers library (von Platen et al.) as the frozen teacher. Consistent with prior observations Haque et al. (2023); Chen et al. (2024b), the teacher's classifier-free guidance (CFG) scales for both prompt and input image have a significant effect on the *degree of edit intensity*—a factor that is often subjective and a matter of personal taste. For most edits we adopt the default $S_T = 7.5$ for the prompt and $S_I = 1.5$ for the input image, with adjustments detailed appendix B. We perform distillation over 40 student timesteps ($\Delta\tau = \frac{1}{40}$), with $k = 50$ updates per step. Optimization is done with AdamW (Loshchilov & Hutter, 2017), using a maximum learning rate of $1 \times 10^{-4}$ after 200 iterations of linear warm-up, followed by cosine decay down to $5 \times 10^{-5}$. This yields just over 2000 distillation iterations per experiment, which take about 40 minutes on a single NVIDIA H200 GPU.

### A.1  TEACHER FORWARD SCHEDULE

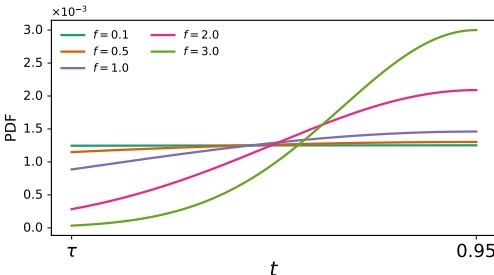

Figure 3: Teacher timestep schedule for different skewness factors $f$.

We employ a stochastic schedule for the teacher forward process, sampling timesteps as

$$t \sim \text{TruncNorm}\big(\mu = b, \ \sigma = (b - \tau)/f, \ a = \tau, \ b = 0.95\big),$$

where $\tau$ is the current student timestep and $f$ controls the skewness of the distribution. Larger $f$ concentrates probability near $b$, making the teacher more likely to operate at higher noise levels. The stochasticity ensures the teacher provides strong gradients every few iterations, which we find effective for avoiding collapse to poor local minima. Figure 13 illustrates how different $f$ values shape the probability distribution. In practice, we use $f = 0.5$, yielding an approximately uniform distribution over $[\tau, 0.95]$.

## B  EVALUATION SCENES AND EDITS

We detail in Table 3 the edits used in our evaluations, applied to the standard Face, Bear, and Person scenes from the Instruct-NeRF2NeRF dataset Haque et al. (2023). The *Edit Prompt* is the editing instruction provided as input to the evaluated methods, while the *Original Prompt* and *Edited Prompt* are employed for CLIP-based evaluation. For each edit, we also report the teacher's text and image CFG scales, $s_T$ and $s_I$, used in quantitative evaluation. Edits with bolded prompts indicate those selected for the ablation experiments.

| Scene | Original Prompt | Edit Prompt | Edited Prompt | Text CFG | Image CFG |
|---|---|---|---|---|---|
| Face | "A man with curly hair in a grey jacket" | "Give him a Venetian mask" | "A man with curly hair in a grey jacket with a Venetian mask" | 7.5 | 1.5 |
| | | "Turn him into a vampire" | "A vampire with curly hair" | 7.5 | 1.5 |
| | | **"Turn him into Tolkien Elf"** | "A Tolkien Elf with curly hair" | 9.0 | 1.5 |
| | | "Turn him into batman" | "A batman" | 7.5 | 1.5 |
| | | **"Turn his face into a skull"** | "A man with a skull head in a grey jacket" | 7.5 | 1.5 |
| | | "Turn him into Albert Einstein" | "Albert Einstein with curly hair" | 7.5 | 1.5 |
| | | "Turn it to a Van Gogh painting" | "A Van Gogh painting of a man with curly hair in a jacket" | 7.5 | 1.5 |
| | | "Give him face paint" | "A man with curly hair in a grey jacket with face paint" | 7.5 | 1.5 |
| Bear | "A stone bear in a garden" | **"Turn the bear to a panda bear"** | "A panda bear in a garden" | 6.0 | 1.5 |
| | | "Turn the bear to a polar bear" | "A polar bear in a garden" | 6.0 | 1.5 |
| | | **"Turn the bear to a grizzly bear"** | "A grizzly bear in a garden" | 5.5 | 1.5 |
| | | "Turn the bear to a wooden bear" | "A wooden bear in a garden" | 8.5 | 1.5 |
| Person | "A man standing next to a wall wearing a blue T-shirt and brown pants" | "Turn him into Iron Man" | "An Iron Man standing next to a wall" | 7.5 | 1.5 |
| | | "Turn the man into a robot" | "A robot standing next to a wall" | 5.5 | 1.8 |
| | | "Make him in a suit" | "A man standing next to a wall wearing a suit" | 6.5 | 1.8 |
| | | **"Turn him into a clown"** | "A clown standing next to a wall" | 6.0 | 1.8 |
| | | "Make him into a marble statue" | "A marble statue of a man next to a wall" | 7.5 | 1.5 |
| | | "Turn him into a cowboy with a hat" | "A cowboy wearing a hat standing next to a wall" | 6.0 | 1.5 |
| | | "Turn him into a soldier" | "A soldier standing next to a wall" | 7.5 | 1.5 |
| | | **"Turn him into a knight"** | "A knight standing next to a wall" | 6.0 | 1.5 |

Table 3: Prompts and CFG values for each edit used for quantitative evaluation.

## C  LIMITATIONS OF INSTRUCT-NERF2NERF IN SPARSE-VIEW SETTINGS

In the sparse-view regime, Instruct-NeRF2NeRF (I-N2N) Haque et al. (2023) fails to produce coherent results. Its underlying Nerfacto (Tancik et al., 2023) model, trained with default configurations for 30K iterations, struggles to reconstruct the scene accurately, generating severe floater artifacts even when rendering the original input poses. These distortions fall far outside the distribution expected by the 2D editor, rendering the resulting edits unusable. Figure 4 illustrates two representative examples of such failures, corresponding to the *Clown* and *Face Paint* edits.

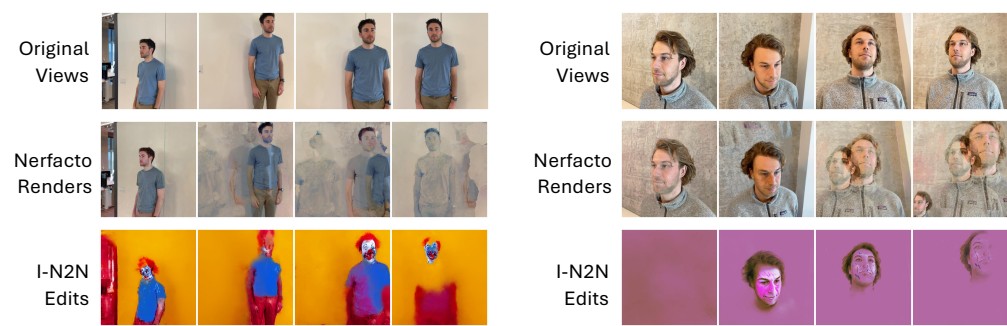

Figure 4: Examples for I-N2N failures in the sparse-view setting.

## D    STUDENT AND TEACHER LIMITATIONS

Figure 5 illustrates the limitations of the student (SEVA) and teacher (Instruct-Pix2Pix) when used individually. Even on the unedited scene, SEVA can struggle to produce high-quality results with only a single input frame, as shown in the second row. When used as an editing baseline—receiving a single edited frame and asked to generate the remaining views—it fails to produce coherent frames (third row). Individual predictions from the teacher (final row) are independent across views, resulting in inconsistent and sometimes implausible edits.

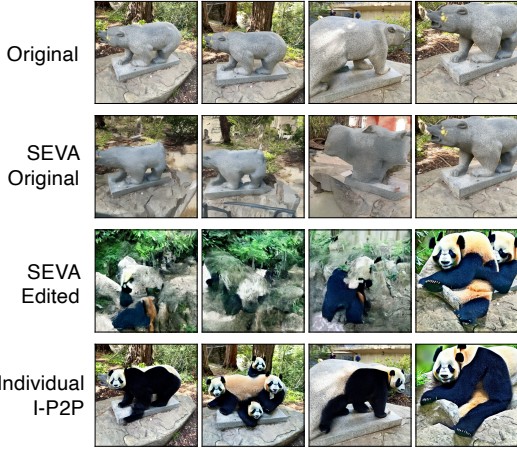

Figure 5: Student and Teacher models limitation example, on the *Bear* scene and *Panda* edit.

## E    EXTENDED QUALITATIVE COMPARISONS WITH BASELINES

We present additional qualitative comparisons to prior methods, including both enlarged versions of the edits shown in Figure 2 and additional edits. Matching red or purple rectangles highlight regions with multi-view inconsistencies.

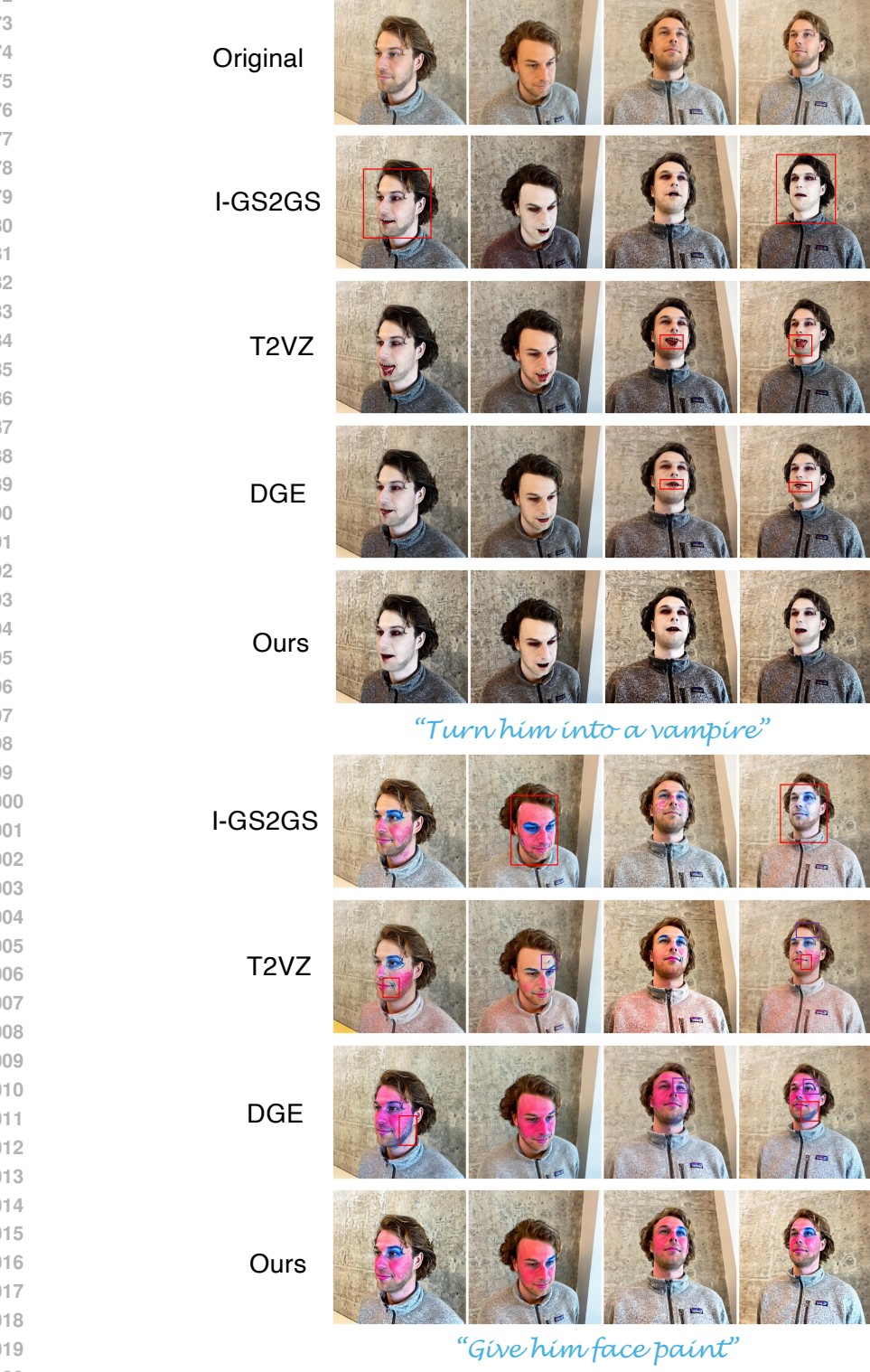

Figure 6: Comparison to baselines on Face scene edits.

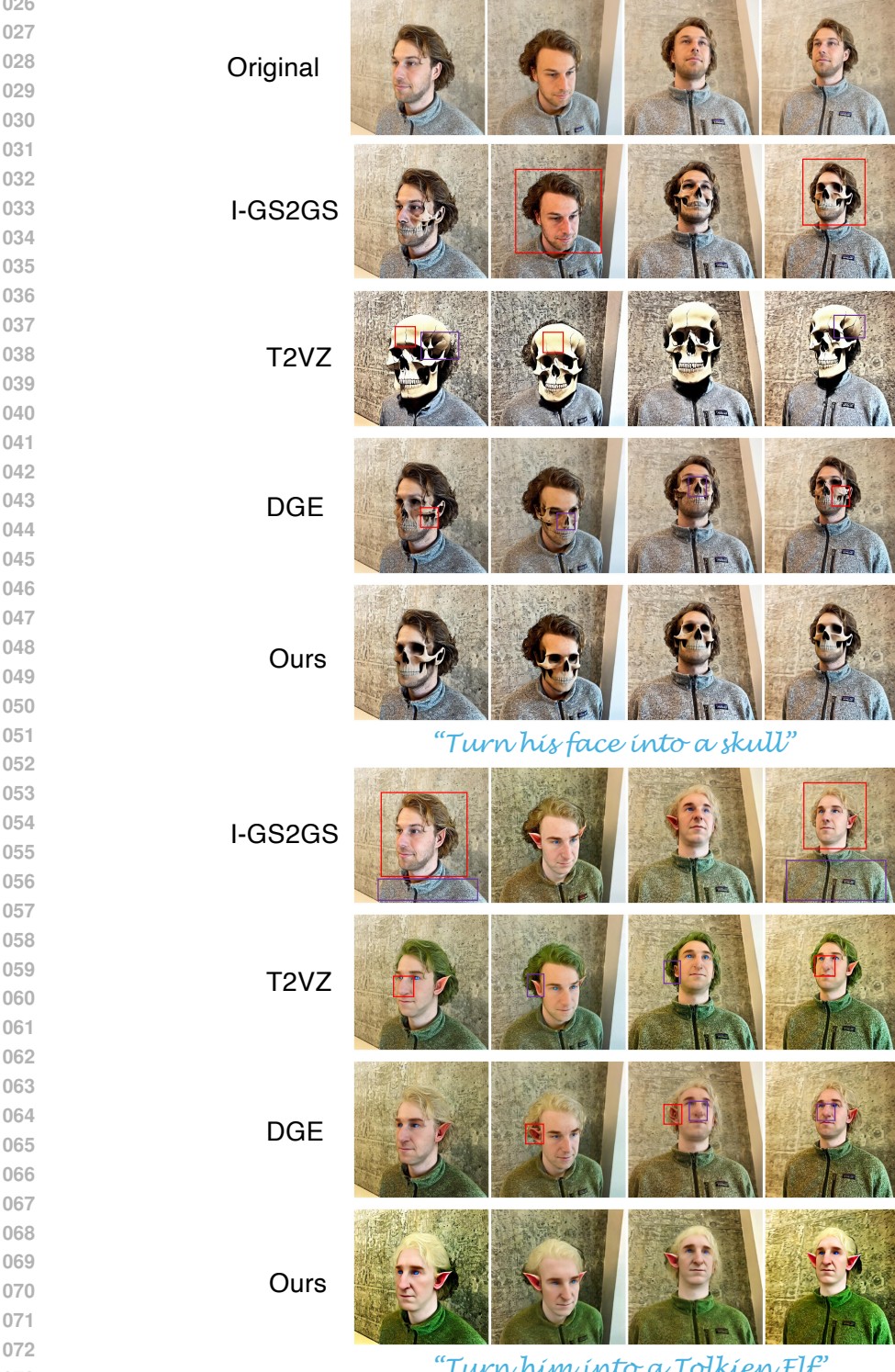

Figure 7: Comparison to baselines on Face scene edits.

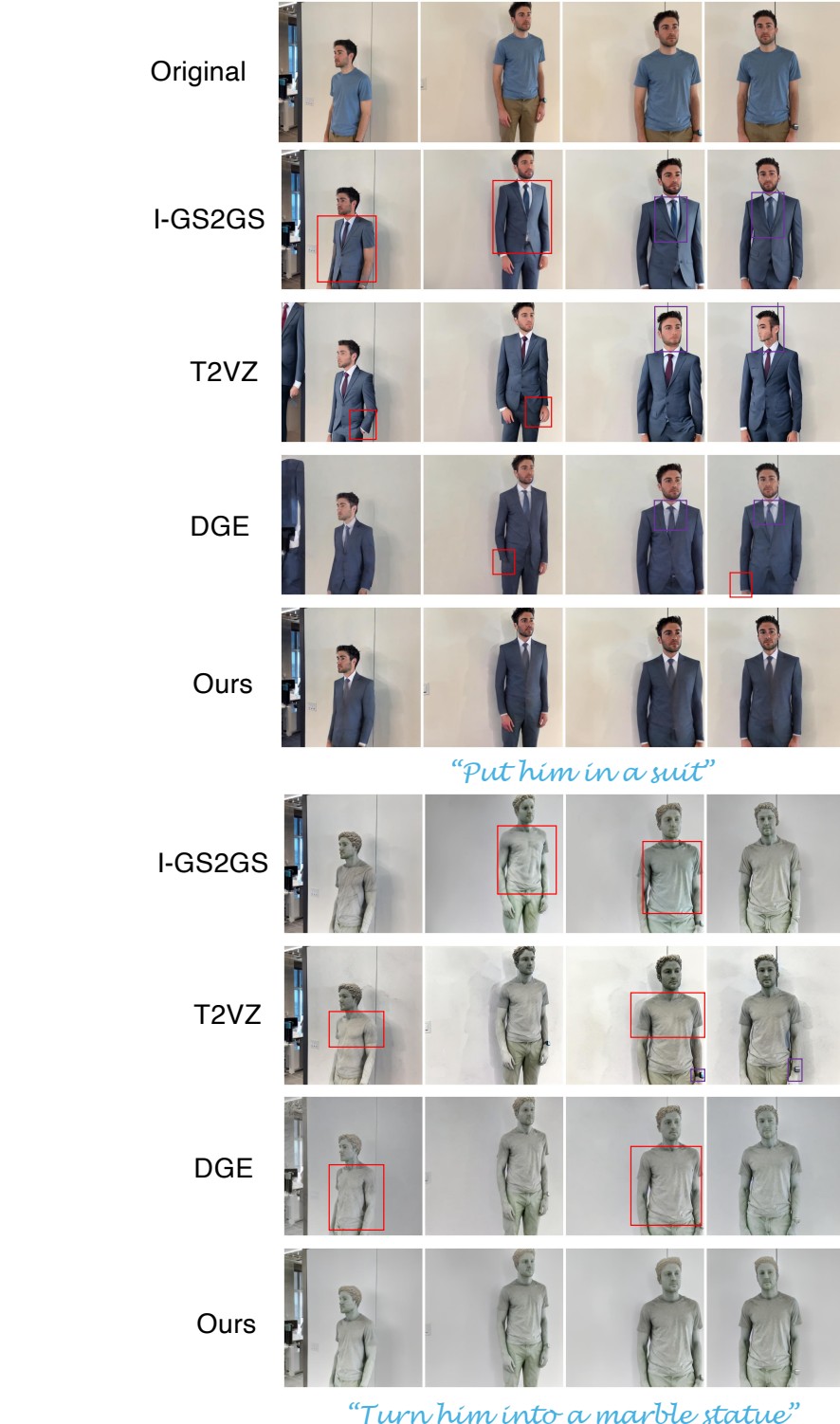

Figure 8: Comparison to baselines on Person scene edits.

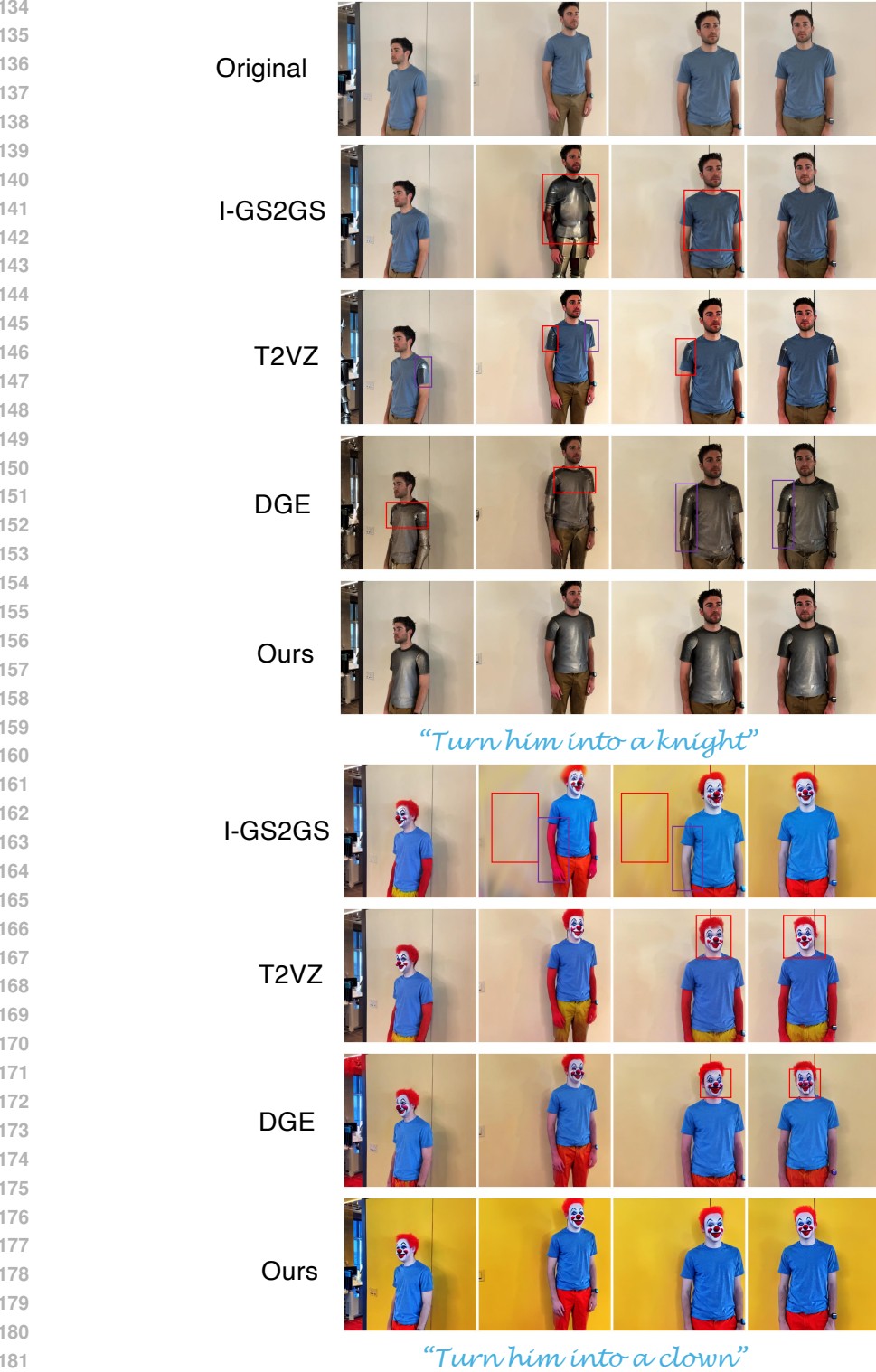

Figure 9: Comparison to baselines on Person scene edits.

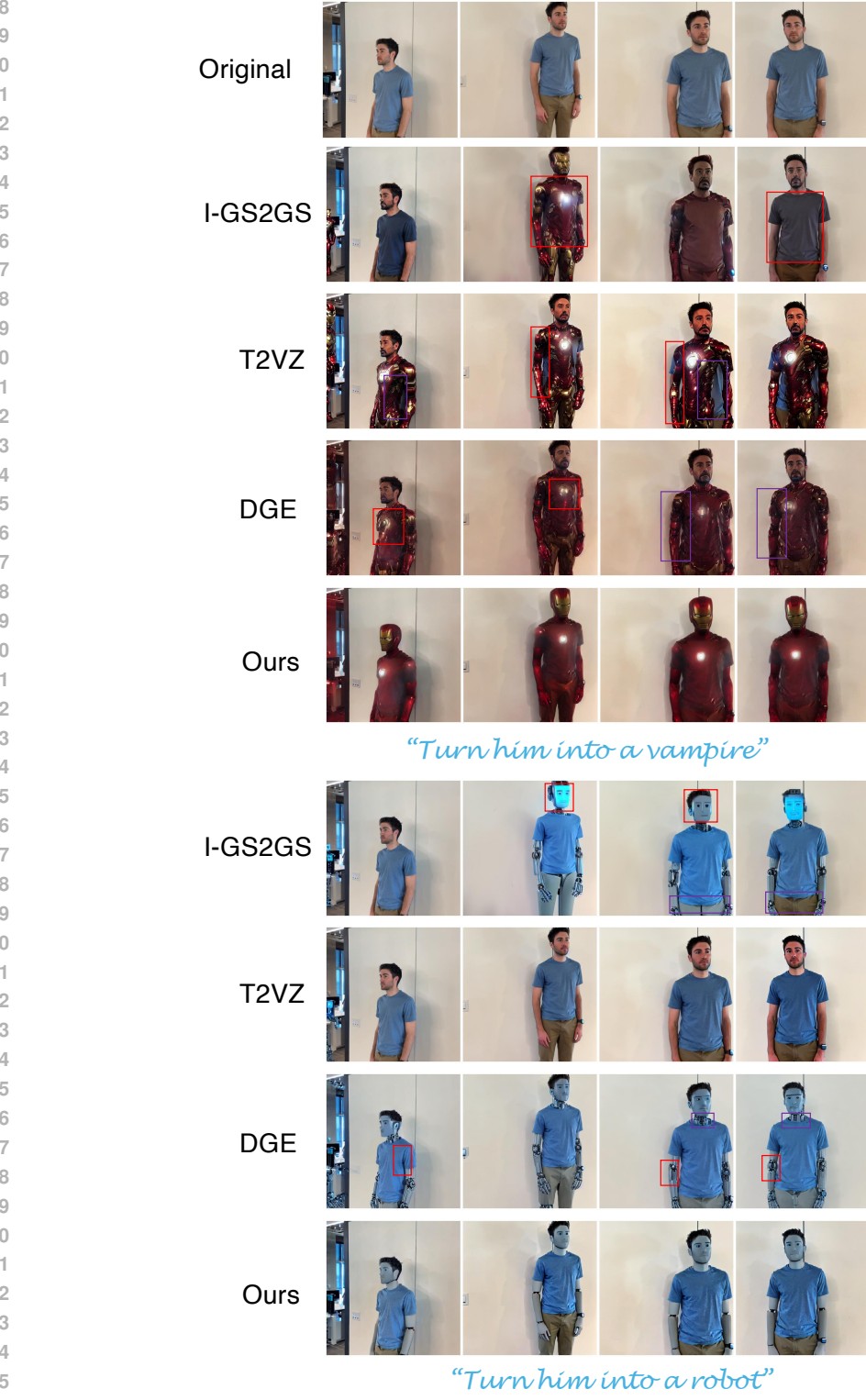

Figure 10: Comparison to baselines on Person scene edits.

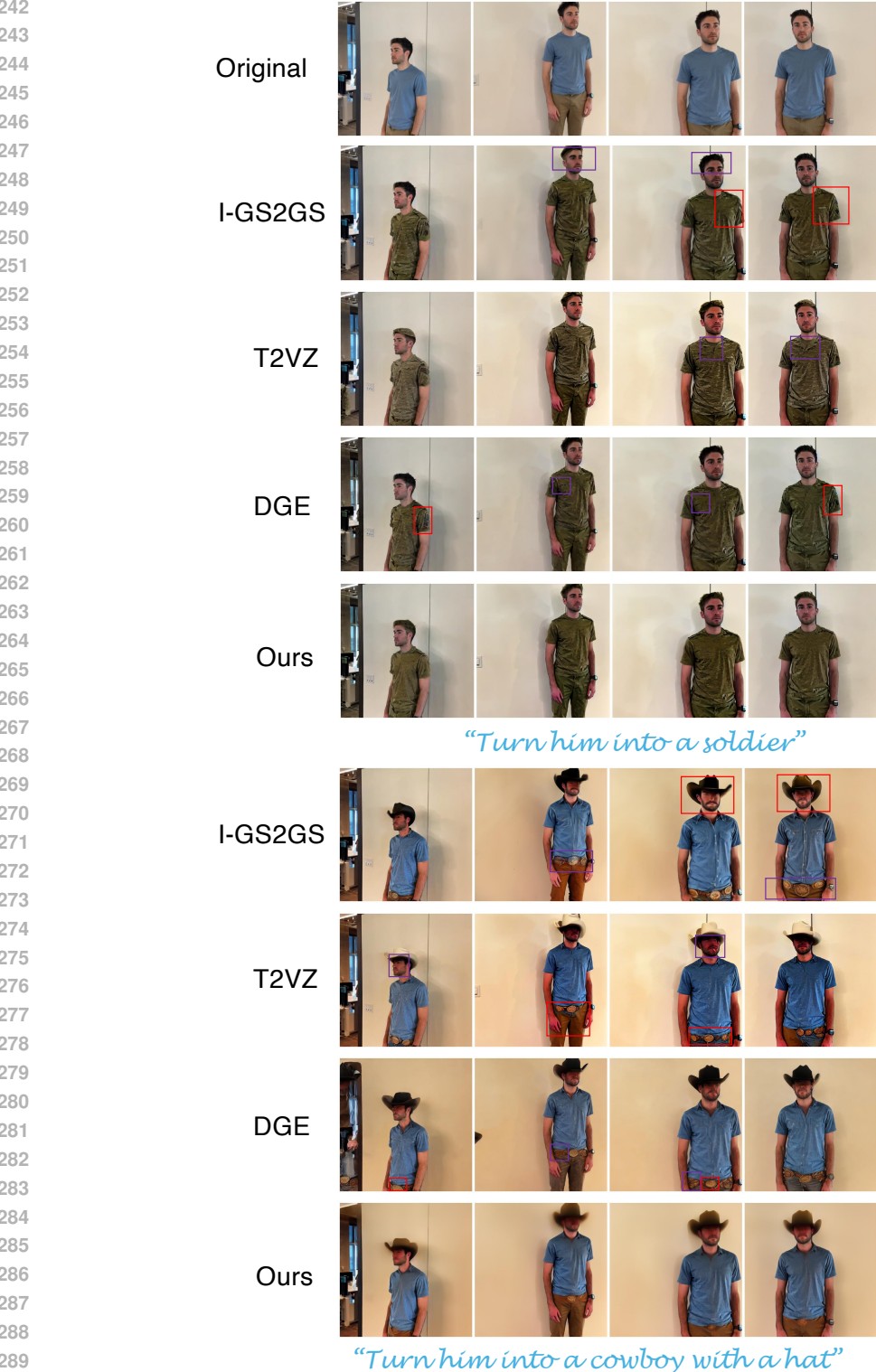

Figure 11: Comparison to baselines on Person scene edits.

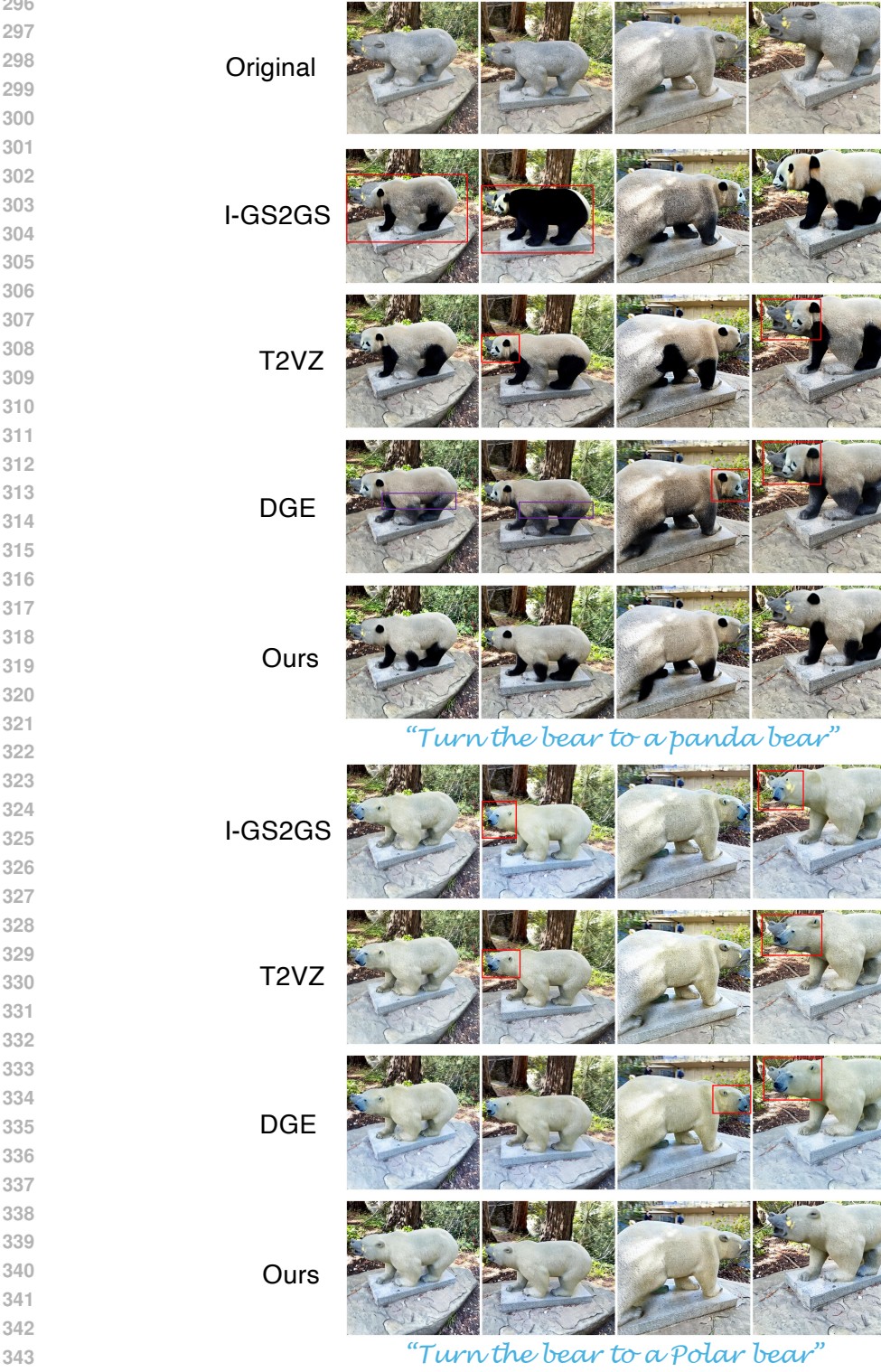

Figure 12: Comparison to baselines on Bear scene edits.

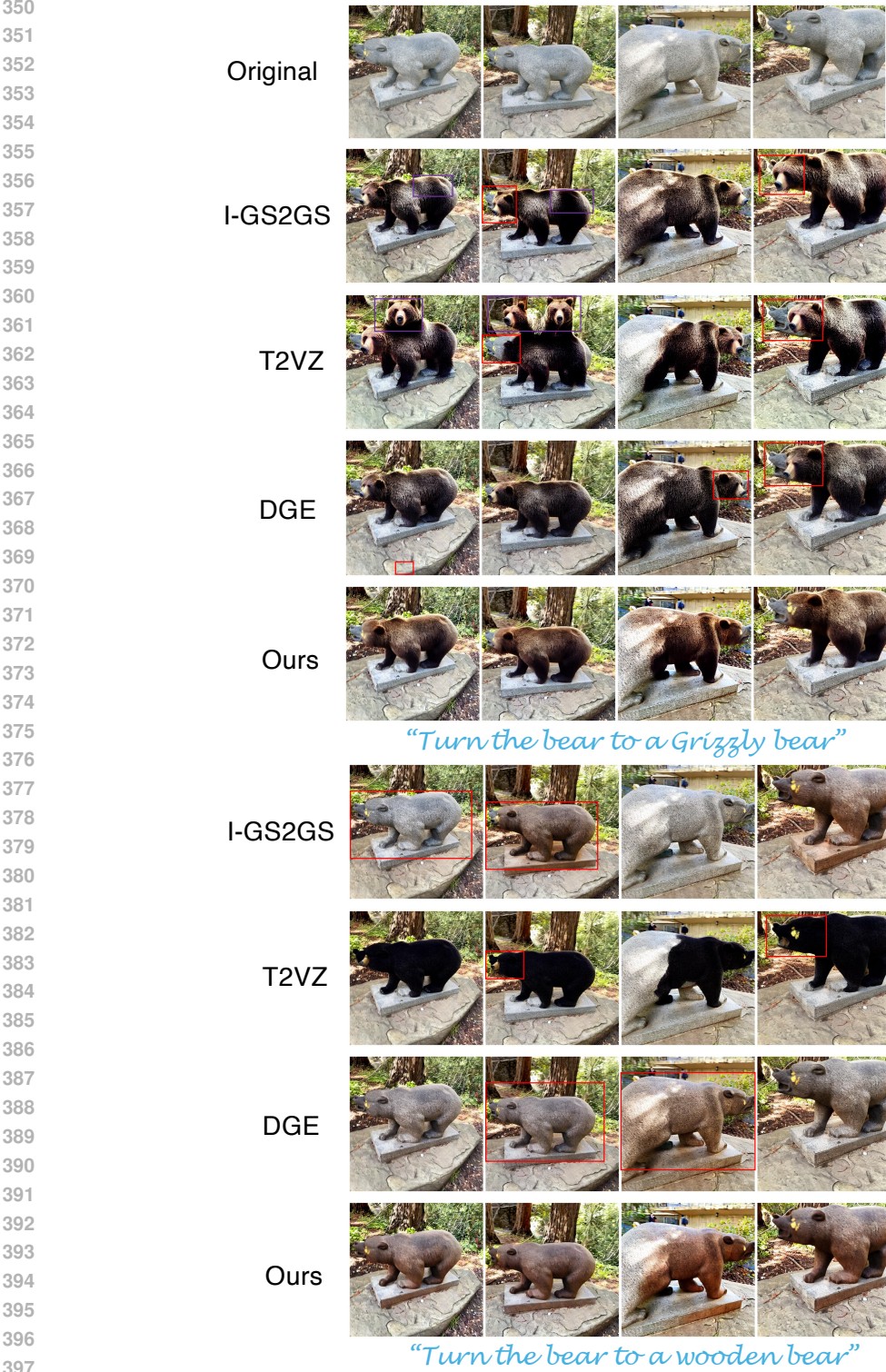

Figure 13: Comparison to baselines on Bear scene edits.

## F    ADDITIONAL RESULTS ON DIVERSE SCENES

In Figures 14, 15 we present further qualitative results of I-Mix2Mix applied to four different scenes: *Car* from the CO3D dataset Reizenstein et al. (2021), *Garden* from the Mip-NeRF 360 dataset Barron et al. (2022), and *Horse* and *Ignatius* from the Tanks and Temples dataset Knapitsch et al. (2017).

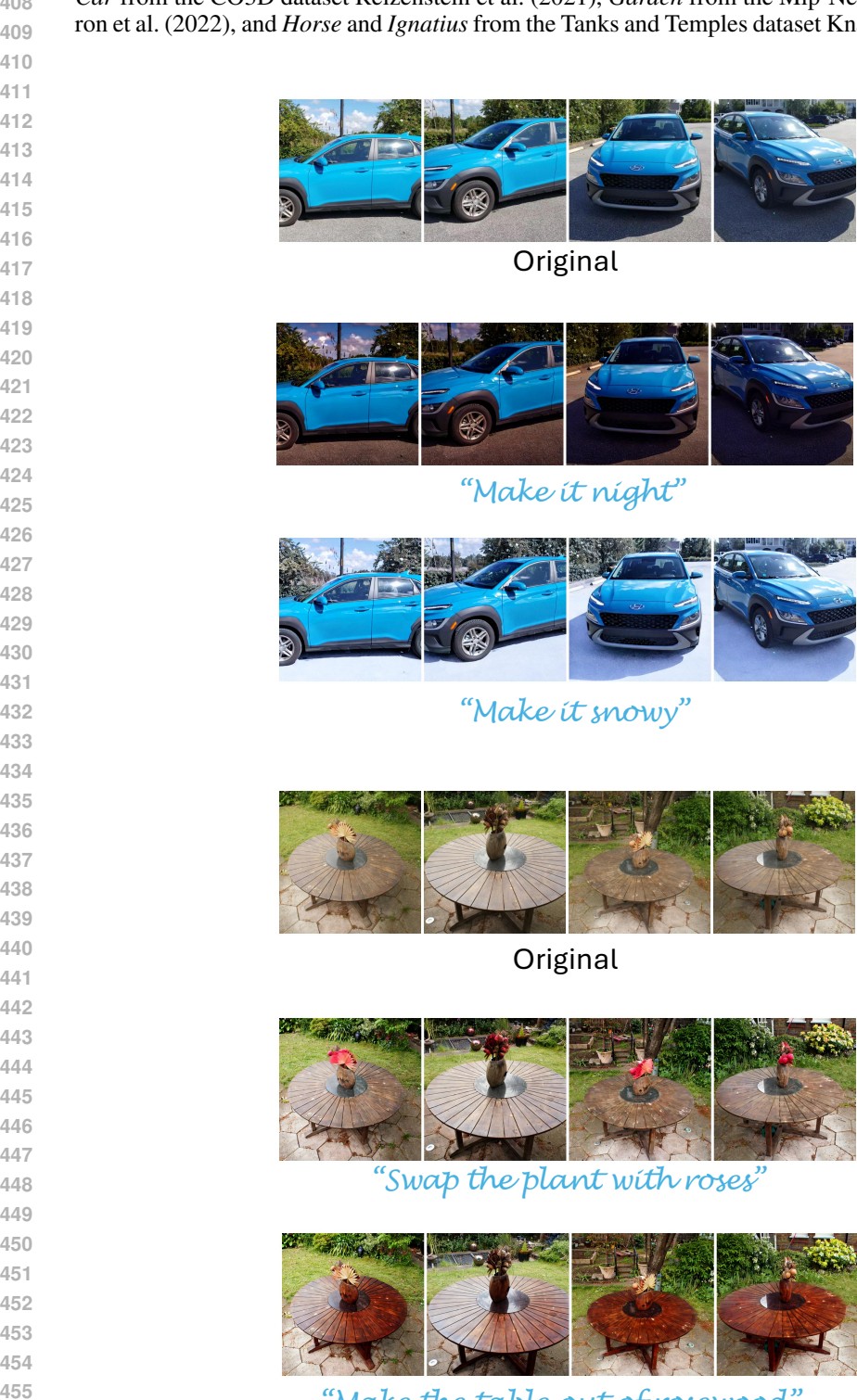

Figure 14: I-Mix2Mix edits on the Car (top three rows) and Garden (bottom rows) scenes.

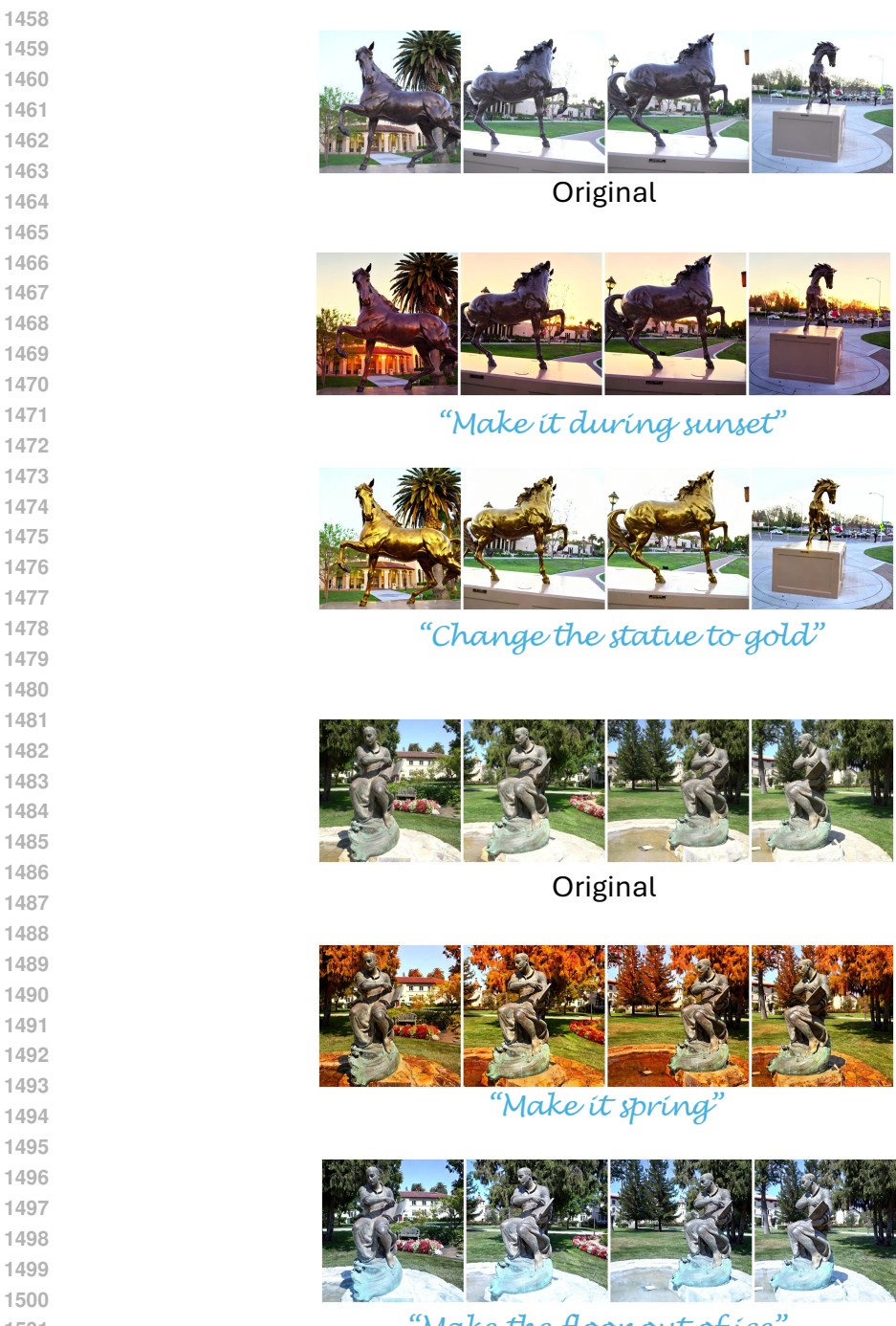

Figure 15: I-Mix2Mix edits on the Horse (top three rows) and Ignatius (bottom rows) scenes.

## G    RESULTS WITH MORE INPUT FRAMES

Figure 16 presents outputs of I-Mix2Mix when using $N = 8$ input frames.

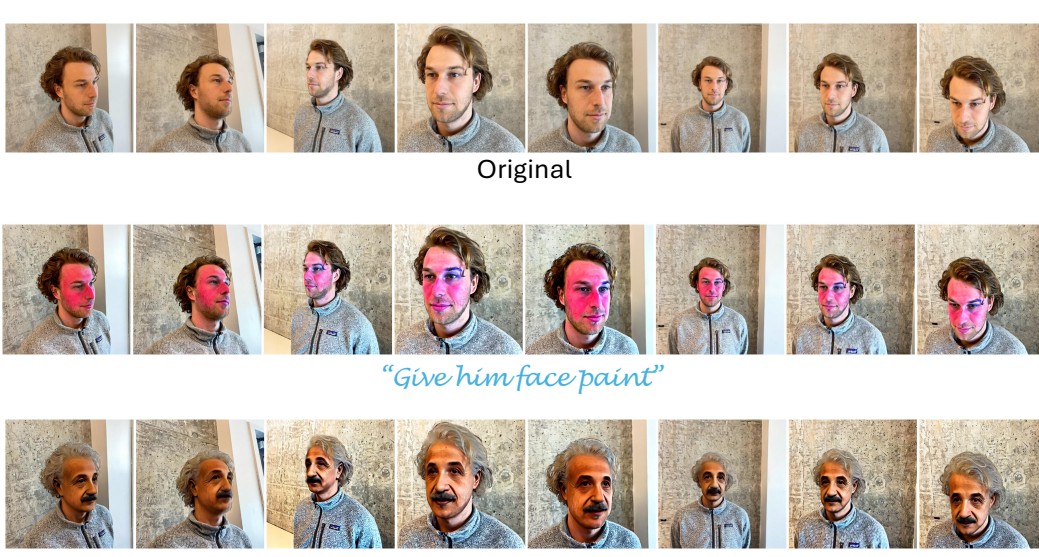

Original

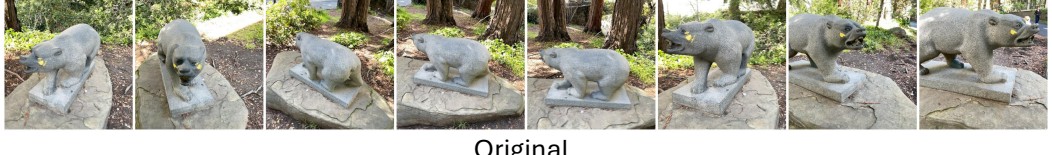

*"Give him face paint"*

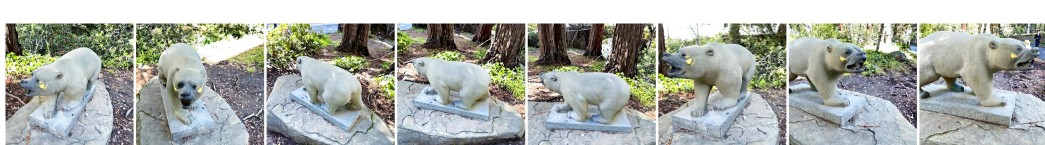

*"Turn him into Albert Einstein"*

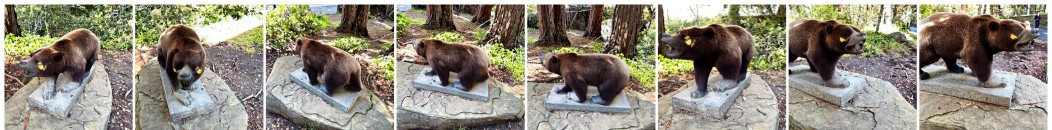

Original

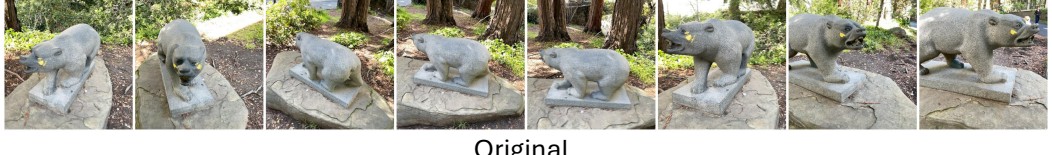

*"Turn him  into a Polar bear"*

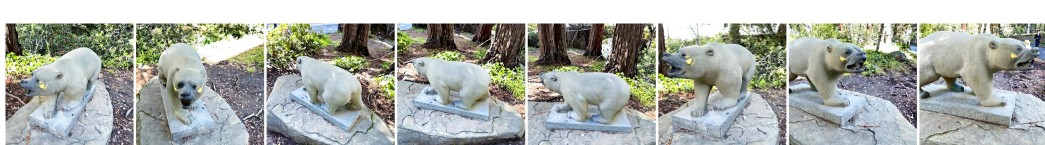

*"Turn him into a Grizzly bear"*

Figure 16: I-Mix2Mix edits on 8 input frames on Face and Bear scenes.

## H    BEYOND IMAGE EDITING

I-Mix2Mix is not tied to a specific editor or to editing tasks, and can in principle generalize to other multi-view conditional generation scenarios. To illustrate this, we used pre-trained ControlNets (Zhang et al., 2023) as teachers to translate multiple depth or Canny maps of a 3D scene into consistent RGB images. Figure 17 shows examples. While outputs respect the conditioning and maintain multi-view consistency, they often appear overly blurry, highlighting limitations of SDS-based optimization (Poole et al., 2022).

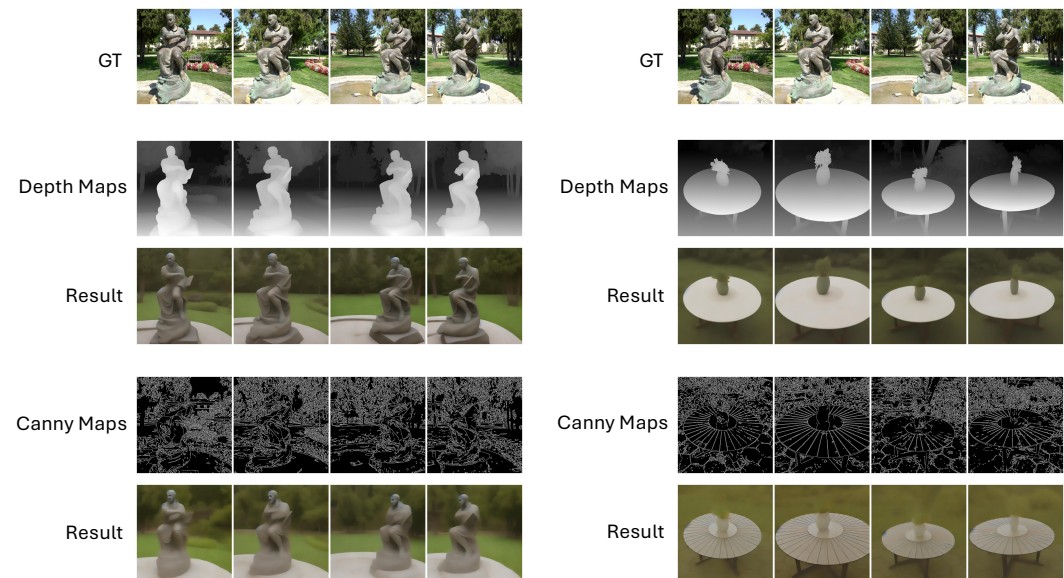

Figure 17: Example results of I-Mix2Mix with Canny edge map and Depth maps as input, with corresponding ControlNet teachers.

# I   USE OF LARGE LANGUAGE MODELS

Large language models were employed as general-purpose assistants for both writing and coding throughout this work.

