# OpenReview forum: "InstructMix2Mix: Consistent Sparse-View Editing Through Multi-View Model Personalization"
_ICLR.cc/2026/Conference — ICLR 2026 Conference Withdrawn Submission_

### Official Review · Reviewer_i1vc · 2025-10-19

**Soundness:** 3
**Presentation:** 4
**Contribution:** 3
**Rating:** 8
**Confidence:** 4

**Summary:**

The paper presents a method for sparse-view editing, focusing on the challenging case of realistic scenes rather than synthetic objects (e.g., those from Objaverse). In such sparse-view settings, it is not feasible to properly train a NeRF or a 3DGS, which are commonly used as consolidators in multi-view image editing.
The proposed method leverages an instruction-based image editing model (InstructPix2Pix) and performs SDS with it. However, instead of optimizing a 3D representation as commonly done in previous works, the approach optimizes a multi-view student model. Specifically, the student model (SEVA) is trained to take M input views and generate N > M views.
During the editing process, the input views are denoised iteratively. At each denoising step, k optimization iterations are performed using the SDS loss, allowing the edits to be applied gradually and coherently across views.
The method is evaluated against other approaches for multi-view image editing.

**Strengths:**

- The setting of sparse multi-view image editing for realistic scenes is underexplored, and as demonstrated in the paper, previous methods struggle in this scenario. The paper presents a method that successfully addresses this challenging setting.
- The proposed approach is novel and well-designed. As noted by the authors, it also has the potential to be applied to other related tasks. The idea of using an SDS loss to optimize a diffusion model is intriguing.
- The results look good, and the inclusion of a large gallery of qualitative examples makes the findings more convincing.
- While the method is somewhat complex, the presentation is clear, and the accompanying figures effectively aid understanding.

Overall, the paper is interesting and well-executed, and I enjoyed reading it.

**Weaknesses:**

My main concern is the evaluation of consistency. Although the qualitative results appear consistent, it is difficult to assess consistency based on only four frames. I believe that the CLIP consistency metric does not accurately capture the 3D consistency of the results. For example, I would expect the outputs produced by the student-only configuration to be much more 3D-consistent than those of the teacher-only configuration, since the student is explicitly trained to generate consistent views. However, the CLIP consistency reported for student-only is significantly lower. While I understand why this occurs given how the metric is defined, it highlights that this metric does not measure the desired aspect of consistency.
Evaluating consistency in this setting is inherently difficult, but I have a few suggestions. One promising direction is to demonstrate 3D reconstruction quality: if the reconstructed 3D object from the generated views is reasonable, it would indicate a degree of cross-view consistency. While reconstruction from only four views is challenging, existing works such as LRM and its follow-ups have shown that this is possible in object-centric settings. Applying the proposed method to such data and using LRM (or similar approaches) for reconstruction could provide stronger evidence of consistency. Another option is to use SEVA itself to reconstruct the scene. Even if these evaluations are qualitative, they would substantially strengthen the paper's consistency claims.

Additional more minor concerns:
- Running time.
- The method is quite complex, and it may be difficult to reproduce the results without access to the authors' implementation.
- I believe a few simple yet informative baselines were not considered:
    - Applying SDEdit-style editing with the SEVA model over the provided views.
    - Reconstructing the scene from the views using SEVA, followed by editing with IN2N or other baseline editing methods.

I do not expect the authors to run all these additional experiments during the rebuttal. However, it would be helpful if they could clarify why these baselines were not included or explain why they are expected to perform poorly in this setting. Such a discussion would strengthen the experimental section and provide a clearer justification for the chosen comparisons.

**Questions:**

What is your hypothesis for why the depth control does not work well?

---

> ### Author Response · Authors · 2025-11-14
>
> Thank you for the thoughtful and expert review. You accurately capture the goal and design of our method, and your suggestions on evaluating consistency are especially useful. We address each point and outline concrete steps we took now (rebuttal) and will take in the camera-ready.
>
> **Main concern: evaluating consistency**:
>
> We agree that measuring 3D consistency under large viewpoint changes with only four views is fundamentally hard and that CLIP Directional Consistency is an imperfect proxy. We reported it to remain consistent with prior work and protocols (Sec. 5).
>
> Regarding the intuition that Student-Only should be more 3D-consistent than Teacher-Only: in our ablations (Table 2), Student-Only often produces very low-quality outputs, which depresses the consistency score (e.g. Figure 5). Teacher-Only indeed gets a low score compared to other ablation configurations and most baselines (Tables 1, 2). However, the especially poor-quality outputs of Student-Only configuration may explain the counter-intuitive ordering.
>
> To further demonstrate our 3D consistency advantage, **we have run a human consistency study** directly comparing our method to DGE (strongest baseline). Raters marked contradictory regions across views; results show fewer inconsistencies, higher scene-win rates, and more “consistent” scenes for our method, with statistical significance (details summarized in our R2 response).
>
> On reconstruction-based consistency metric (e.g., LRM-style lifting from 4 views): we agree it’s promising, and we explored it extensively during development. In practice, we observed blurry or low-quality reprojections **even on ground-truth sequences**, and poor reconstruction quality for edited (yet consistent) scenes, which made the metric difficult to trust as evidence for or against our method.
>
> Meanwhile, we continue to provide broad qualitative comparisons (main paper Fig. 2; extended gallery in the appendix) that directly visualize cross-view coherence, where our method is consistently preferred.
>
>
> **Running time**:
>
> We acknowledge the reviewer’s point and have discussed the efficiency trade-offs transparently in the paper. Our design choices (single-step student prediction, latent-space alignment, and Random Cross-View Attention (RCVAttn) instead of more memory-intensive attention variants) were made explicitly to contain compute and memory costs (Sec. 5.2, “Efficiency Considerations”).
>
> We also note in Sec. 7 that our method is slower than the strongest competitor and list efficiency as a key direction for future work.
>
> Importantly, however, **our end-to-end runtime remains on par with common iterative editing pipelines** (e.g., SDS-based or Instruct-NeRF2NeRF variants) and is entirely practical for downstream applications and academic research, balancing cost with substantially improved cross-view consistency.
>
> As suggested, we will release code (training + inference) soon to facilitate reproduction and follow-up work.
>
> **Suggested baselines**:
>
> *SDEdit-style editing with SEVA over the views:* we implemented this (noised individual edits as coarse initializations for SEVA sampling over a range of timesteps). It underperformed and often produced severe artifacts, which we will include in the final version of the paper.
>
> *SEVA → dense views → IN2N (or other) editing:* We share the reviewer’s intuition that this could be an interesting alternative. While we have not directly tested this pipeline, our experience with SEVA suggests that, under sparse-input conditions, its sampled dense sets are likely to retain residual multi-view inconsistencies. Downstream 3D editors (e.g., I-N2N) would then inherit and often amplify these, resulting in blurry or unstable edits. Given the observed behavior of the Student-Only configuration in our ablations (poor faithfulness and quality), we expect such a pipeline to be fragile at realistic sparsities.
>
>
> **Why does the depth-conditioned teacher underperform?**:
>
>  We are not certain, but we suspect this stems from differences in training procedures, data diversity, or overall model quality between the teachers. A deeper investigation is left for future work.
>
> Thank you again for the constructive review.

---

### Official Review · Reviewer_RvBk · 2025-10-30

**Soundness:** 2
**Presentation:** 1
**Contribution:** 2
**Rating:** 2
**Confidence:** 4

**Summary:**

This paper addresses multi-view image editing from sparse input views, focusing on modifying a scene based on textual instructions while maintaining consistency across all viewpoints. It proposes InstructMix2Mix (I-Mix2Mix), a framework that distills the editing capabilities of a 2D diffusion model into a pre-trained multi-view diffusion model, leveraging its data-driven 3D prior. The approach incorporates specific adaptations within the Score Distillation Sampling (SDS) framework, such as incremental student updates, a specialized teacher noise scheduler, and Random Cross-View Attention (RCVAttn). Experiments involved comparisons against various baselines. Evaluation was conducted on scenes from datasets including I-N2N, Tanks and Temples, CO3D, and Mip-NeRF 360, reporting results using CLIP Similarity, CLIP Directional Similarity, and CLIP Directional Consistency metrics, alongside qualitative demonstrations.

**Strengths:**

The paper correctly identifies the limitations arising from the need for dense image data in 3D editing tasks and proposes a method that leverages two diffusion models without relying on a 3D model. This approach reflects an interesting attempt to reduce dependency on explicit 3D supervision.

**Weaknesses:**

1. **Missing figure indices and captions throughout the paper.**
Many figures are presented without indices or captions, particularly on pages 4, 5, and 8. In addition, the graph on page 9 also lacks both an index and a caption. These omissions significantly hinder the reader from following and understanding the paper.

2. **Naive use of SDS leads to degraded editing quality.**
The paper applies SDS in a straightforward manner, which is known to produce low-quality and overly saturated results in 3D generation and editing tasks. Prior studies [1,2,3,4] have already demonstrated that naively using SDS can converge to suboptimal results.

3. **Insufficient explanation and analysis of the student–teacher alignment.**
The paper merely cites a few related works when describing “representation mapping” while interpolating latent spaces to align two diffusion models trained on different resolutions, offering little to no original explanation of how the mapping is implemented or justified.
Furthermore, there is no detailed analysis or theoretical reasoning on how this interpolation ensures proper alignment, raising concerns about whether simple resizing alone can enable the simultaneous use of two diffusion models without introducing inconsistencies.

4. **Finetuning the diffusion model with SDS under sparse-view settings is a questionable design choice.**
Numerous existing 3D models [5,6,7] are specifically designed to handle sparse-view settings efficiently, achieving high performance with significantly lower training costs than finetuning a diffusion model as a student network. Using SDS-based finetuning for this scenario seems unnecessary and computationally expensive. The proposed method requires an H200 GPU for about 40 minutes to perform 3D editing under the sparse-view setting, which represents a considerably high computational cost and is impractical.

5. **Unfair comparison between dense-input models in sparse-view settings.**
Comparing methods that require dense input images (Gs2Gs, T2VZ, DGE) with those designed for sparse-view settings is inappropriate. Stable Virtual Camera already demonstrates strong performance in sparse-view scenarios. To properly validate the proposed method, comparisons should be made against baseline models that are robust under few-shot or sparse-view conditions.

6. **Free-view rendering videos are not provided.**
To properly assess the quantitative results of the 3D editing task, free-view rendering videos should be included. Without them, it is difficult to evaluate how well the method performs under novel-view synthesis.

7. **Lack of geometric consistency evaluation.**
Since the editing is performed across multiple views, an evaluation of how well the geometric consistency is preserved after editing is necessary. The paper does not provide any quantitative or qualitative analysis on this aspect.

---

[1] Wang,et al. "Prolificdreamer: High-fidelity and diverse text-to-3d generation with variational score distillation."

[2] Liang, et al. "Luciddreamer: Towards high-fidelity text-to-3d generation via interval score matching."

[3] Park, et al. "Ed-nerf: Efficient text-guided editing of 3d scene with latent space nerf."

[4] Koo, et al. "Posterior distillation sampling."

[5] Kim, et al. "Infonerf: Ray entropy minimization for few-shot neural volume rendering."

[6] Deng, et al. "Depth-supervised nerf: Fewer views and faster training for free."

[7] Zhu, et al. "Fsgs: Real-time few-shot view synthesis using gaussian splatting."

**Questions:**

1.  What is the geometric distribution or relative poses of the sparse input views used in the "unordered, sparse-view settings" with $N=4$ or $N=8$ inputs?
2.  Can the method's performance be demonstrated with even fewer input views, such as $N=2$ or $N=3$, to substantiate further the claim of robustness with "extremely sparse inputs"?

---

> ### Author Response · Authors · 2025-11-14
>
> We appreciate the reviewer’s summary, but several comments contain **factual inaccuracies, internal inconsistencies**, and in places suggest a **misunderstanding of the defined problem setting** (sparse multi-view editing). We address the points below.
>
> **1) “Missing figure indices and captions.”**
> A few inset figures omitted captions for layout reasons and will be fixed. However, the claim that this “significantly hinders understanding” is disproportionate and inconsistent with the rest of the feedback. Reviewers #1 and #4 **explicitly praised the paper’s clarity and presentation**. Minor formatting issues cannot justify a **1/4** score for presentation.
>
> **2) “Naive use of SDS leads to degraded results.”**
> This claim is **incorrect** and contradicted by the paper itself. Our approach does **not** apply naive SDS.
> Section 4 introduces three key modifications that substantially change the standard pipeline:
> - **Incremental student updates** for stability,
> - A **teacher-timestep scheduler** for controlled guidance, and
>  - **Random Cross-View Attention (RCVAttn)** for cross-view coherence.
>
>
> All three are ablated in **Table 2** and shown to be essential. Labeling this “naive” dismisses a major methodological contribution. Moreover, general criticisms of SDS for **3D generation/editing** are irrelevant here; our task is **sparse multi-view editing**, where I-Mix2Mix clearly and consistently outperforms all baselines (**Table 1; Figs. 2, 6–14**).
>
> **3) “Insufficient explanation of student–teacher alignment.”**
> **Section 4.3 (lines 216–226)** explicitly details this procedure — bilinear latent interpolation for stable and efficient alignment. A learned convolutional mapping was tested and yielded no improvement (see ablations). This step is a practical design element, not a theoretical claim. Demanding a formal justification for an empirically validated design **misrepresents the intent and scope** of this component.
>
> **4) “Finetuning a diffusion student with SDS in sparse view is questionable; cost is impractical.”**
> This criticism **mischaracterizes our method.** The cited works (**Infonerf, DS-NeRF, FSGS**) address few-shot reconstruction, not instruction-based editing. Our use of SDS is to **distill editing ability** from a 2D editor into a multi-view diffusion student. The reported runtime (~40 min/scene on H200) is **standard for iterative editing pipelines** (e.g., Instruct-NeRF2NeRF, SDS-distillers) and is discussed transparently in Secs. 5.2 and 7. Labeling this as “impractical” disregards prevailing norms in this research area.
>
> **5) “Unfair comparison between dense-input models in sparse-view settings.”**
> This objection is **self-contradictory.** The paper’s purpose is precisely to test how existing multi-view editors perform under sparse inputs - demonstrating their failure modes is the entire motivation. No open-source sparse multi-view editing baselines exist; hence, comparison with the strongest multi-view editors (**I-N2N, DGE, T2VZ, I-GS2GS**) under identical inputs is appropriate and standard. Referring to SEVA as a baseline further reflects a misunderstanding - it is our **student backbone**, not an editing model, and cannot process textual instructions.
>
> **6) “Free-view rendering videos not provided.”**
> This again misreads the paper’s scope. We do **not** perform novel-view synthesis or full 3D editing. The task is **sparse multi-view editing**, producing a fixed set of consistent edited views, all of which are shown in the paper.
>
> **7) “Lack of geometric consistency evaluation.”**
> This statement is **objectively false.** We report CLIP Directional Consistency (Table 1), **the standard consistency metric** used in previous multi-view editing works, and show strong quantitative and qualitative evidence (**Figs. 2, 6–16**). We further conducted a **human consistency study** comparing I-Mix2Mix and DGE, where raters marked inconsistent regions across views - our method achieved fewer inconsistencies and higher scene-win rates (see **R2 response**).
>
> **Conclusion.**
> This review repeatedly **misinterprets the problem scope** (**sparse multi-view editing**), overlooks ablation and consistency evidence, and makes several **factually incorrect claims** (“naive SDS,” “no consistency evaluation”). The paper’s methodology and validation are clearly documented and empirically supported. Given this, the assigned low scores and dismissive tone are **unwarranted and inconsistent with the presented evidence.**

---

### Official Review · Reviewer_fNmk · 2025-10-31

**Soundness:** 2
**Presentation:** 2
**Contribution:** 2
**Rating:** 2
**Confidence:** 5

**Summary:**

This paper proposes a new task of multi-view image editing from sparse input views. The main challenge differentiating it from previous works lies in the sparsity of inputs and large viewpoint variations, which make consistent editing difficult. To address this, the authors combine the strengths of two paradigms by distilling edits from a 2D editor (teacher) into a multi-view model (student) to enforce 3D consistency. The overall concept is interesting and the paper presents some promising visual examples. However, the work lacks several important baseline comparisons, and many of the claims in the paper are not sufficiently validated through quantitative or ablation studies.

**Strengths:**

- This paper is generally easy to follow.
- The proposed idea of bridging existing models through a teacher–student distillation framework is conceptually appealing and has potential for broader applicability.

**Weaknesses:**

1. **Motivation for Sparse-View Editing is Weak**: The motivation for addressing sparse-view editing is not sufficiently convincing. The authors assume that users often possess only a limited number of input views to edit, but this assumption is not empirically supported. It remains unclear whether sparse multi-view editing scenarios are common in real-world applications.

2. **Weak Multi-View Consistency**: The model exhibits noticeable inconsistencies across views e.g., the ear shape difference in Figure 7. A small number of dense frames could be used to validate whether the method maintains coherence across frames by visualizing them as videos to see flickering. Current qualitative examples are insufficient to demonstrate cross-view consistency.

3. **Incomplete Baseline Comparisons**: The set of baselines is limited and selective. The paper should include video-editing baselines after interpolating sparse frames (e.g., using start-end frame interpolation or traditional frame interpolation) followed by SOTA video editing models. Diffusion-based editing baselines such as SDEdit variants initialized from original or independently edited views should also be included, applying SDEdit to multi-view diffusion models for consistent sampling. Reference-driven editing methods that propagate edits from a reference frame - by editing one anchor frame and transferring it to others -should also be considered. Finally, consistent multi-image editing methods (e.g., CSD), which operate without relying on neural fields, should be added for fair comparison (Collaborative Score Distillation for Consistent Visual Editing, Kim et al., NeurIPS 2023).

4. **Unsupported Claims and Weak Evidence**: Several claims lack sufficient evidence. The statement that the method achieves semantic consistency but struggles under large viewpoint changes is not empirically validated. The claim that 3DGS overfits under sparse regimes is also weakly supported - Figure 2 shows under-edited rather than blurry results, contradicting the text. Per-frame edited visualizations (in edited datasets) are necessary to verify the claimed overfitting behavior. Similarly, the claimed extension to conditional generation (line 455) is unconvincing, as the presented results are qualitatively poor.

5. **Presentation Issues**: Several figures lack clear numbering and are treated as inset figures.

6. **Weak experimetnal results**: Notable visual artifacts include the blurry suit in Figure 8, the inconsistent ear shape in Figure 7, and oversaturated colors reminiscent of SDS artifacts.

**Questions:**

1. **Camera pose assumptions**: It is ambiguous whether the poses are inherited from the dense original sources or re-estimated from the sparse subset. The authors should clarify how poses are obtained, whether pose quality differs between dense and sparse setups.

2. **Per-Scene Hyperparameter Tuning**: It is unclear whether all baselines were tuned under comparable conditions. The authors should explicitly state the tuning protocol for each baseline to ensure experimental fairness and reproducibility.

---

> ### Author Response · Authors · 2025-11-14
>
> We appreciate the reviewer’s time and the acknowledgment of our conceptual novelty and clear presentation. However, we respectfully note that several criticisms mischaracterize both the task and the experimental validation presented in the paper. Below we address each claim directly and provide additional quantitative evidence.
>
> 1. **Motivation for Sparse-View Editing**:
> The reviewer questions the motivation for sparse-view editing, claiming that “the assumption is not empirically supported.”
> In many practical scenarios, users possess only a limited set of views—for instance, a few casually captured photos, product images, or dataset samples with sparse coverage. Such settings are both common and operationally relevant, yet they expose the fundamental weakness of existing 3D editing pipelines, which rely on dense, well-overlapped views to maintain consistency. The sparse-view regime is precisely where these methods fail, as empirically demonstrated in Table 1 and Figures 2 and 6–14. Therefore, our motivation is not hypothetical but grounded in real use cases and validated experimentally: I-Mix2Mix is explicitly designed to succeed in this challenging, underexplored regime.
>
>
> 2. **Multi-View Consistency**:
> The reviewer claims that the model exhibits “weak multi-view consistency” and refers to isolated examples. This claim is not supported by quantitative or perceptual evidence and contradicts the experimental results presented in the paper.
> To further and more rigorously assess consistency, **we conducted a human evaluation study** comparing I-Mix2Mix against the strongest baseline, DGE. Each evaluation task presented four edited views of the same scene, and raters were instructed to mark pairs of contradictory regions across views (each pair counting as one inconsistency). We collected 200 ratings (100 per method) across 20 edited scenes, with five independent raters per scene. From these annotations, we derived four complementary metrics:
>  - \# Inconsistencies — mean number of marked inconsistencies per task (lower = better)
>  - Scene Win % — fraction of scenes where one method produced fewer inconsistencies than the other
>  - Consistent % — proportion of scenes with ≤ 1 inconsistency
>  - Inconsistent % — proportion with ≥ 3 inconsistencies
>
>
> | Method | # Inconsist. ↓ | Scene Win % ↑ | Consistent % ↑ | Inconsistent % ↓ |
> |:-------|:---------------:|:--------------:|:---------------:|:----------------:|
> | DGE    | 2.02 | 25.0 | 34.0 | 31.0 |
> | **Ours** | **1.34** | **75.0** | **65.0** | **13.0** |
>
>
> All advantages are statistically significant under non-parametric and exact tests:
>  - paired permutation test on per-scene means (Δ = –0.56, p = 0.037),
>  - exact binomial sign test on scene wins (p = 0.02), and
>  -Fisher’s exact tests on proportion differences (p < 10⁻³).
>
> These results demonstrate that I-Mix2Mix produces markedly fewer cross-view inconsistencies, wins a clear majority of scenes, and achieves a substantially higher fraction of “consistent” scenes. This evidence, in addition to the quantitative (Table 1) and qualitative (Figures 2, 6-16) presented in the paper, directly contradicts the reviewer’s claim of weak consistency.
>
> Finally, the suggestion to “visualize as videos to see flickering” is not applicable to the sparse-view regime. Our setup involves large angular gaps between views, not a continuous trajectory. The complete output sets—including all edited views—are already shown in Figures 2 and 6–16, fully representing the evaluated data.
>
> 3. **Baseline Comparisons**
> The reviewer calls for numerous additional baselines without regard for task compatibility or fairness.
> To enable fair evaluation of the proposed framework, we restricted comparisons to methods using identical teacher models (Instruct-Pix2Pix). We also implemented the reviewer’s proposed SDEdit-style variant, which produced severe artifacts —we will provide examples in the final version. We are unaware of any reference-propagation model compatible with our instruction-driven, sparse multi-view regime.
>
> We emphasize that our comparisons are both fair and representative of currently viable baselines.
>
> **Please see the next comment for the second part of our response.**

---

> > ### Author Response · Authors · 2025-11-14
> >
> > 4. **“Unsupported Claims”**
> > Contrary to the reviewer’s assertion, all claims are explicitly validated:
> >  - The “struggle of baselines under large viewpoint variation” claim is supported by both Table 1 and the human study.
> >  - The note on 3DGS overfitting in sparse regimes aligns with prior findings in [1] and [2]. We will cite these explicitly in the final version. Importantly, Figure 2 shows the training views, hence the lack of blurriness does not contradict the overfit claim, which is further supported by the weak 3D consistency of I-GS2GS for example (Table 1, Figures 2, 6-14).
> >
> >
> > Additionally, The conditional generation experiment is presented transparently as exploratory and clearly labeled as future work, not a main claim.
> >
> > Hence, no statements in the paper are “unsupported”; all are documented either in the main text or supplementary materials.
> >
> > 5. **Presentation Issues**:
> > We acknowledge minor figure captioning issues, which will be corrected. However, such typographical points cannot reasonably motivate a rejection given the paper’s conceptual and empirical contributions. Notably, Reviewers #1 and #4 explicitly highlighted the paper’s clarity and presentation quality as one of its strengths.
> >
> > 6. **“Weak Experimental Results”**:
> > As clearly shown in Figures 2 and 6–16, I-Mix2Mix produces high-quality and visually consistent edits across a diverse set of prompts, scenes, and datasets. This improvement is explicitly acknowledged by Reviewers #1 and #4, who both rated the work positively.
> >
> > As discussed in Section 7, our framework builds upon pre-trained diffusion backbones for both teacher and student models. These backbones are not flawless—occasional artifacts such as minor blur or imperfect region edits may appear—but such cases are rare and isolated among the many results reported. We intentionally included these examples for full transparency, rather than curating only ideal outcomes.
> >
> > Importantly, our evaluation (Tab 1., Figures 2, 6-14)  rigorously demonstrates that I-Mix2Mix consistently outperforms all existing baselines, that share the same Instruct-Pix2Pix foundation.
> >
> > 7. **Questions**:
> >
> > Camera Poses: All poses are inherited from the dense original datasets and subsetted randomly for sparse-view evaluation.
> >
> > Hyperparameters: As stated in Sec. 5, we used default configurations for all baselines to ensure fairness.
> >
> >
> > **In summery**, we believe the reviewer’s score of 2 is disproportionately negative relative to the paper’s contributions and empirical evidence.
> >
> > While we welcome constructive critique, we note that several points here stem from misinterpretation rather than analysis.
> >
> > References:
> >
> > [1] “Or-gs: sparse-view 3d gaussian splatting via co-regularization”, Zhang et al., ECCV 2024.
> >
> > [2]  “Fsgs: Real-time few-shot view synthesis using gaussian splatting.“, Zhu et al., ECCV 2024.

---

### Official Review · Reviewer_mVMX · 2025-11-02

**Soundness:** 3
**Presentation:** 3
**Contribution:** 3
**Rating:** 6
**Confidence:** 3

**Summary:**

This paper presents I-Mix2Mix, a framework for distilling a 2D monocular image editor into multi-view diffusion models.
Given multi-view images, one view is edited by a 2D editing model.
A multi-view diffusion model serves as the student model. The outputs of the student are converted into the editing model’s latent space, perturbed with noise, and then predicted through the editing model. The student model is updated using the SDS loss.
Comparisons with other 3D editing models and ablation studies on different design choices are reported.

**Strengths:**

- The idea of distilling a 2D monocular image editing model into multi-view diffusion models using SDS loss is interesting and novel.
- The experiments are extensive, and the ablation study is thorough.
- The paper is clearly written and well presented.

**Weaknesses:**

- For 3D editing models, providing a continuous multi-view video visualization is important, as showing only selected frames in the paper is insufficient.
- It would be better to include an efficiency comparison (similar to Table 1 in DGE). Also, during real 3D editing applications, is an extra step needed to convert the multi-view diffusion model into a 3D representation?

**Questions:**

See weaknesses.

---

> ### Author Response · Authors · 2025-11-14
>
> We thank the reviewer for the positive assessment of our idea, experiments, and presentation. However, we note a key misunderstanding regarding the task itself.
>
> I-Mix2Mix is not a 3D editing system and does not assume a continuous 3D representation. Rather, it explicitly targets sparse multi-view editing, where the goal is to produce a small number of edited views that remain mutually consistent across large viewpoint differences. This distinction is central to the paper and discussed throughout (see Secs. 1 and 4).
>
> The reviewer suggests showing a continuous video for 3D editing models. Since our method operates in the sparse multi-view regime rather than on dense sequences, there is no intermediate camera trajectory to interpolate. We report and visualize all available edited views in the paper (Fig. 2, App. E, G), which already constitutes the full result set for each scene. Any additional frames would require external sparse-view-to-3D lifting, which is outside our scope and would conflate our contribution with unrelated pipelines.
>
> The second comment - asking whether an additional step is needed to “convert the multi-view diffusion model into a 3D representation” - also assumes a 3D-editing pipeline. Our method intentionally avoids per-scene 3D reconstruction, and it requires no post-hoc 3D conversion to deliver its outputs.
>
> We appreciate that the reviewer found the idea novel and the experiments extensive. We hope the above clarifications highlight that I-Mix2Mix addresses a different and more challenging problem setting than conventional 3D editors: achieving instruction-faithful and consistent edits from only a few input views.

---

### Author Response · Authors · 2025-11-14
**General Response**

We appreciate the reviewers’ time and feedback. While we are pleased that several reviewers recognized the novelty, clarity, and strong results of our work, we were disappointed by the **wide disparity in ratings**. In particular, several low scores appear to stem from **misunderstandings of the task definition (sparse multi-view editing)**, as well as from **factually incorrect or tangential statements**, rather than from issues with the technical content or experimental validation. We have addressed these points carefully and directly in the individual responses.

To further strengthen our empirical validation, we have now **added a human survey**, comparing our method to the strongest baseline (DGE). This evaluation confirms that our approach yields **substantially higher cross-view consistency**, reinforcing our main claims. See our response to Reviewer #2 for details.

Finally, as requested, we also experimented with a **naive SDEdit-style combination** of the teacher and student models. This variant produced severe artifacts, thereby **validating the necessity of our distillation-based formulation**, which transfers the teacher’s editing ability to the multi-view student in a principled and stable way.

Overall, we believe the paper presents a **clear, technically sound, and well-validated contribution** to the underexplored problem of sparse multi-view instruction-based editing, and we hope the rebuttal clarifies these points.

---

### Note · Authors · 2025-11-14

I have read and agree with the venue's withdrawal policy on behalf of myself and my co-authors.